# STING activation induces cytotoxic and immune responses in meningiomas via inflammatory cell death pathways

Meningiomas are common tumors of the central nervous system that are typically treated with surgery or radiation, but lack established systemic therapies. Activation of the stimulator of interferon genes pathway with an agonist such as 8803 can trigger anti-tumor immune responses. Using integrated molecular approaches, here we show that this pathway is targetable in both neoplastic and immune populations within the meningioma microenvironment. Meningioma tumor cells exhibit promoter hypomethylation and increased chromatin accessibility of the STING genomic locus, associated with robust expression of this gene. Treatment of diverse patient meningiomas ex vivo with 8803 induces direct tumor cytotoxicity through inflammatory cell death pathways, including induction of gasdermin D membrane pore formation. Release of necrotic tumor debris triggered by 8803 activates macrophages and upregulates matrix metalloproteinase production, facilitating degradation of extra-cellular collagen. Injection of preclinical meningiomas with 8803 induces survival benefits, including in an immunocompetent orthotopic setting, through remodeling of the tumor microenvironment, immune infiltration, and downregulation of tumor-mediated immune suppression, thereby nominating 8803 for treatment consideration in meningiomas.

Meningiomas are common tumors of the central nervous system (CNS) that arise from the fibrous covering of the brain and spinal cord[1]. There are three grades of meningiomas designated by the World Health Organization (WHO), which are associated with an increasing likelihood of recurrence. While meningiomas may be treated effectively with surgery and radiotherapy in some cases, a subset of patients will be resistant to these approaches, and the tumor will recur with progressive morbidity or mortality. Transcriptional studies using single-cell sequencing and spatial techniques have revealed a surprising degree of heterogeneity within meningiomas, particularly among high-grade lesions that are more likely to be treatment-resistant[2-6]. Additional molecular approaches, including genetic, transcriptional, and epigenetic profiling, have further identified key biomarkers that predict the risk of recurrence or aggressive behavior[2,4,7-12]. Integrated

analyses of these modalities have consequently led to the emergence of prognostic molecular subgroups associated with distinctive copy number changes, transcriptional signatures, and DNA methylation profiles, with varying levels of immunologic infiltration[2,4,11,13]. Although there is a growing appreciation for the role of the immune system in brain tumor prognosis and clinical decision-making, limited work to date has focused on immune cells in meningiomas. Additional insights into meningioma immunology could spur therapeutic advances and potentially reduce the morbidity of invasive surgical approaches that are ineffective in some patients.

Prior efforts to characterize meningioma immunology have utilized bulk tissue datasets, flow cytometry, or scRNA sequencing approaches to describe cellular constituents and phenotypes within the tumor microenvironment (TME)[2,3,14]. Deconvolution of bulk

✉e-mail: amy.heimberger@northwestern.edu

 1

methylation has characterized broad immunological clusters and high-level cell states, such as naïve or memory, resting or activated, and macrophage polarization status across meningioma grades, and suggested broad immune suppression associated with myeloid cell enrichment, and a low frequency of T cells. Two recent clinical trials evaluated immune checkpoint inhibitors as monotherapy in recurrent meningiomas. Although treatment was well tolerated, only a subset of patients derived a therapeutic benefit[15,16] indicating that the reprogramming of the TME will be essential for the efficacy of immune-based therapies.

An immune modulatory strategy that might leverage a myeloid-dominated microenvironment is the activation of the STING pathway. Activation of STING via cGAS-mediated recognition of double-stranded DNA results in phosphorylation and nuclear localization of IRF3, culminating in the production of type I interferons (IFN) and induction of local immune responses[17]. Agonists of the STING pathway, such as the potent dinucleotide analog 8803[18], are being investigated pre-clinically in the indication of gliomas to induce proinflammatory immune responses[19–23]. As STING expression is limited in neoplastic cell populations of gliomas and other CNS tumors, the mechanism of action for STING agonists has mainly focused on immune-mediated anti-cancer activities. To date, the direct effects of STING agonists on solid cancer cells have not been fully clarified. STING activation has been shown to induce apoptosis[24,25], necroptosis[26,27], and autophagy[28] in leukemia cells; however, cell death pathways have not been well evaluated in solid cancers.

In this study, we utilize integrated transcriptional, epigenetic, and spatial multiplex immunofluorescent approaches to identify the widespread expression of STING in the meningioma TME, including in neoplastic and infiltrating immune cell populations. The STING agonist 8803 induced direct cytotoxicity of ex vivo resected meningiomas, independent of CD45+ populations, in a process that depended in part on programmed necrotic cell death pathways (necroptosis, pyroptosis, ferroptosis) and reactive oxygen species production. Membrane pore formation via GSDMD oligomerization was induced in meningioma cells after 8803 treatment, thereby allowing release of damage-associated molecular patterns (DAMPs) that synergistically activate local immune populations and result in upregulation of collagen-degrading genes. Efficacy of STING targeting was confirmed across multiple preclinical meningioma models, providing a compelling justification for a clinical trial of 8803 in surgically refractory meningiomas.

## Results
### Single-cell and spatial analyses identify collagen-mediated immunosuppression as a predominant feature of the meningioma microenvironment

To characterize the immunobiology of meningiomas and to identify immune therapeutic targets, single-cell RNA-seq (scRNA-seq) datasets with adjacent dura samples[2,3] were analyzed. These data encompassed twenty-two patients and a total of 30,206 immune cells (Fig. S1). Lineage marker analysis revealed fifteen immune (CD45+) populations (Fig. 1A), with distinctive profiles between meningioma and dura. For example, phagocytic (phago) tumor-associated macrophages (TAMs) were significantly enriched in meningiomas relative to the adjacent dura (Fig. 1B, C, gold color; Fig. S2) and exhibited expression of CD163 and HLA-DR (Fig. 1D), as previously reported[29]. These cells localize to meningioma stromal niche regions and express immunosuppressive proteins such as HAVCR2 (TIM3) and LAIR-1 (Fig. 2A), but rarely CD47 (Fig. S3). Because the deposition of collagen is an early embryonic step during the development of the meninges[30] and regulates the immunological microenvironment in other cancers[31,32], additional focused analysis of collagen family members using scRNA-seq and spatial multiplex immunofluorescence was performed. COL1A1, COL3A1, COL4A1, COL5A1/A2, and COL6A1/A2 were the most expressed

collagens in meningioma cells, and these were found to colocalize in concentric laminar patterns across meningioma grades (Fig. 2B, left; Fig. S4A, B). These regions also contained abundant LAIR-1 expression (Fig. 2B, right), consistent with the role of collagens as ligands for this inhibitory immune receptor[33]. Adjacent normal brain did not express COL1A, COL4A, or COL6A (Fig. S4C), and infiltration of macrophages was not identified in the brain or adjacent dura (Fig. S4C, D). An additional enriched immune population in meningiomas were heat-shock protein (HSP) TAMs (Fig. 1B, C), suggesting a particular affinity of macrophage populations to the meningioma microenvironment. HSP proteins are intracellular DAMP molecules released from damaged or dying cells that induce inflammatory responses.

In contrast, terminally exhausted T cells (T.Tex; identified from markers highlighted in Fig. 1A) and NK cells were more frequently observed in meningioma-adjacent dura (Fig. 1B, C), as previously reported[34,35]. Consistent with these observations, myeloid-associated chemokines such as CCL8 and CCL4L2 were upregulated in meningiomas (Fig. 2C, blue), while the dura exhibited higher T cell-associated chemokines (red). T and NK immune effector cells could potentially exert cytotoxic functions based on the expression of GZMA, PRF1, and NKG7; however, they also display markers of exhaustion including TIGIT, LAG3, and PD-1 (Fig. 1D). Three NK states were identified in the meningioma microenvironment from scRNA-seq data based on established gene expression markers for resting, activated, and exhausted phenotypes (Fig. 2D). Exhausted NK cells, based on the co-expression of markers of activation (such as granzyme family members) and inhibition (KLRD1), were more frequent in the meningioma relative to the dura (Fig. 1). This population also exhibited TGF-β-mediated pathway activation (JUND and SMAD7), which is typically not reversible with TGF-β antagonists[36]. Cumulatively, these data indicate that meningiomas are a myeloid-enriched disease, with suppression of cytotoxic immune functions (including NK and T cells) and collagen-mediated immune suppression in tumor tissue relative to the adjacent dura. Given these findings, therapeutic strategies that drive pro-inflammatory immune responses into the meningioma hold the greatest promise for treatment efficacy.

### Epigenetic activation drives STING expression across meningioma subgroups and cell types

Signaling pathway analysis of immune cells indicated that MHC, galectin, CypA, Annexin, CCL, CLEC, TNF, Lck, CD86, and IFN-β were upregulated across multiple populations in the meningioma microenvironment (Fig. S5A). Notably, these genes have been previously associated with the STING pathway[37–43], suggesting a possible immune therapeutic target for meningiomas. To further investigate the expression of STING in meningiomas, the complete set of cells from the meningioma TME was annotated, including myeloid, T/NK, endothelial, fibroblasts, pericytes, meningioma, and extra-cellular matrix (ECM) expression-associated meningioma (ECM meningioma) cells (Fig. 3A). Across these populations the expression of STING (TMEM173) was highest in meningioma cells and endothelium (Fig. 3B), while other immune targets such as CD47 were minimally expressed (Fig. S3) or were only expressed on the ECM meningioma cells (such as CD274/PD-L1 and CD276/B7-H3)(Fig. 3B). Spatial multiplex immunofluorescent staining revealed that STING expression was high in both high-grade and benign meningiomas, but not brain or dura (Fig. S5B, C). More specifically, STING was identified in SSTR2+ meningioma cells, CD163+ macrophages, and CD31+ endothelial cells (Fig. 3C), indicating that this pathway may be targetable across multiple cell types within the meningioma TME. Downstream canonical STING activity was heterogeneous within STING+ tumor cells and macrophages (as denoted by p-IRF3 expression; Fig. 3C), indicating potential for further pathway activation using chemical agonists.

Though therapeutic STING targeting is under active investigation in gliomas, this protein exhibits minimal expression in neoplastic

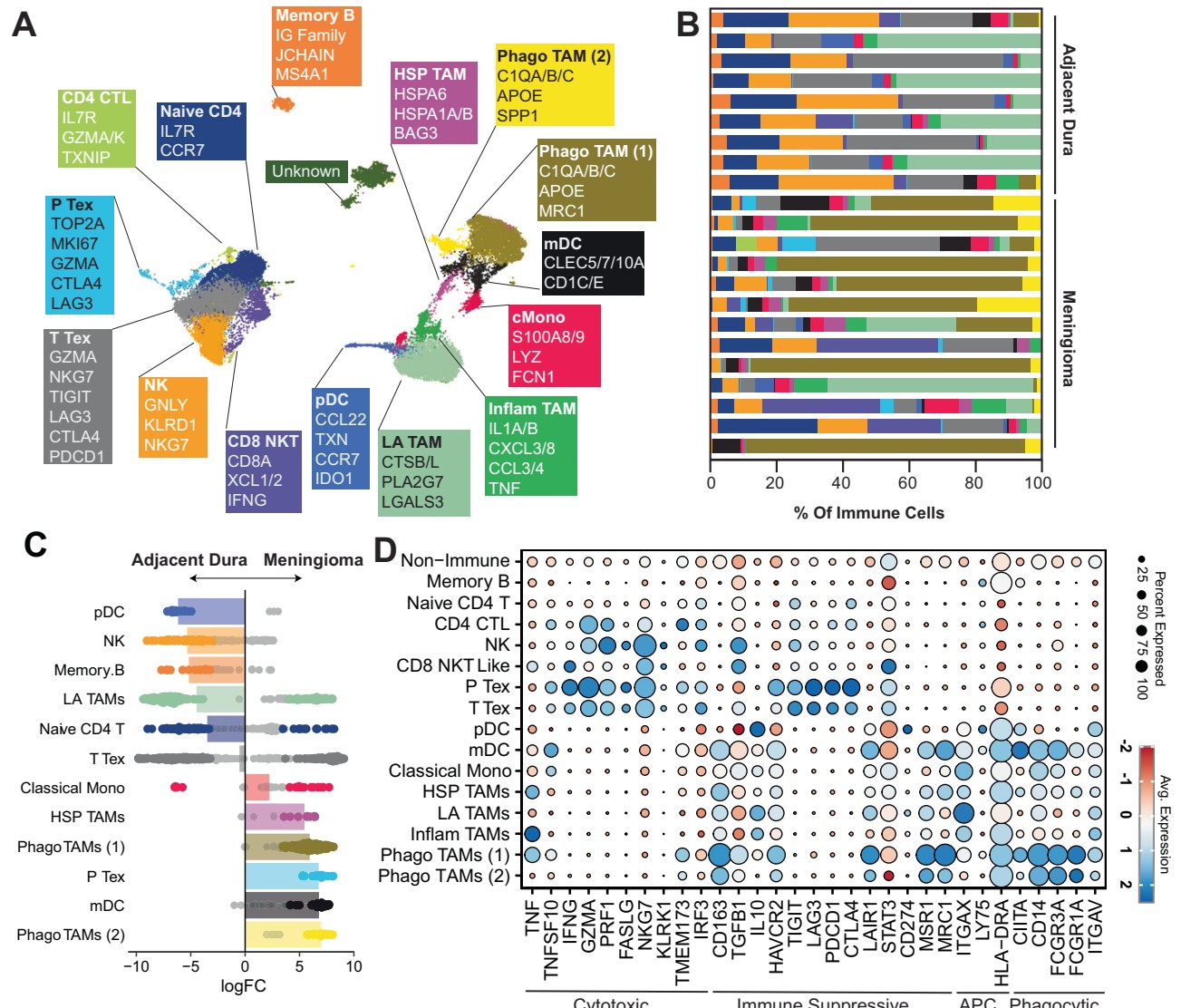

**Fig. 1 | The immune landscape of the meningioma microenvironment is dominated by innate immune suppressive populations. A** Fifteen distinct immune populations (CD45 positive) were identified using single-cell RNA-sequencing data from meningioma (n = 13 patients) and adjacent dura tissues (n = 9 patients) based on the listed markers. Differentially expressed genes (DEGs) were found using standard parameters in Seurat FindAllMarkers (which use default non-parametric Wilcoxon rank sum test on all clusters), and examples are shown in each cluster box[2,3]. 30,206 immune cells were included. **B** Myeloid and lymphoid populations exhibited differential distribution between adjacent dura and meningiomas, such as the enrichment of phagocytic tumor-associated macrophages (phago.TAMs, shown in olive color) in meningioma tissue. **C** T and NK cell populations were present at higher levels in dura, whereas macrophages were dominant within meningiomas. Differential abundance testing was conducted on batch corrected sample level clusters using MiloR standard workflow, including: build-graph, makeNhoods, countcells, calcNhoodDistance, and testNhoods. Visualization was performed using plotDAbeeswarm. Neighborhoods are colored if p-value < 0.05; otherwise, shown in grey. Immune cell populations with fewer than 5 significant neighborhoods were removed from the strip plot. A two-sided negative binomial generalized linear model with spatial FDR correction was used. Data was also analyzed using t-tests (with multiple testing correction) and results are further plotted in Fig. S2 (p < 0.01; q < 0.01). **D** Gene expression dot plot of selected functional markers among the general immune populations shown in (**A**). 30,206 total immune cells were used for analysis. Bubble size corresponds to the percent of cells expressing each marker; colors indicate average expression. Phagocytic TAMs have high expression of M2-like markers such as CD163, CD204, and CD206.

populations of these and other types of CNS tumors[20]. As our data suggests that meningiomas are an exception to this pattern, epigenetic profiling was performed to understand regulatory differences that may drive STING expression. Pseudo-bulk analysis of snATAC-seq data in meningioma (n = 4 patients; 29,100 cells) and glioma (n = 2 patients; 4850 cells) cells was performed to produce lineage-specific ATAC tracks. Separate clustering of cancer-lineage-specific immune populations revealed consistent chromatin accessibility in both tumor types. By contrast, meningioma neoplastic cells exhibited chromatin accessibility on par with myeloid immune lineages, while glioblastoma tumor cells had negligible open chromatin in the STING promoter

(Fig. 3D, blue highlight). Analysis of bulk methylation patterns between tumor types indicated that the STING promoter region was hypo-methylated in meningioma (average β = 0.10, n = 1412 patients) but mostly hypermethylated in glioblastoma (average β = 0.72, n = 170 patients) (Fig. 2D, center). Across a subset of meningioma samples with paired RNA-seq data (n = 584 patients), methylation of the promoter site cg16983159 exhibited the strongest correlation with STING expression (rho = −0.54, p < 0.001; Fig. 3E), consistent with associations in glioma previously reported[20]. We did not observe significant differences in STING expression in meningiomas across prognostic clinical variables, including WHO grade, tumor location, methylation

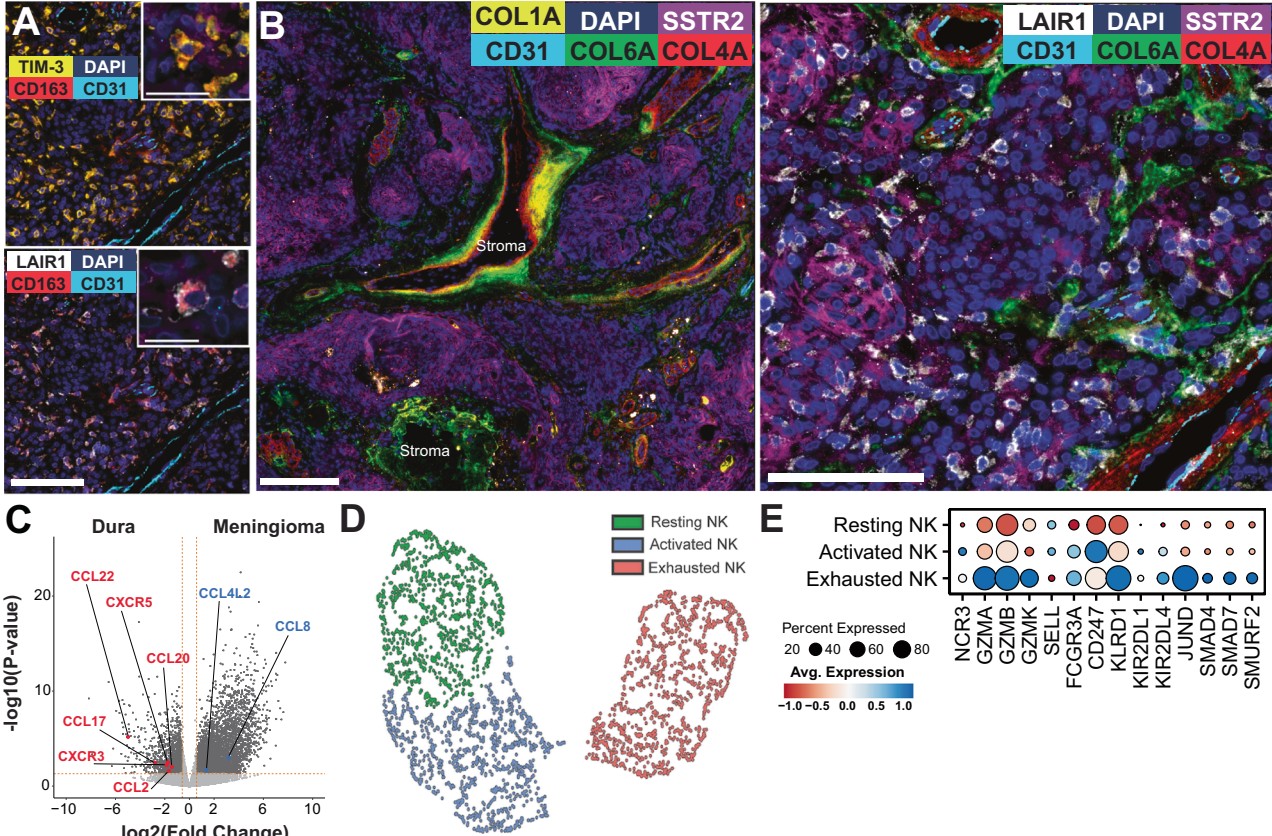

**Fig. 2 | Collagen-mediated immune suppression is present within meningiomas. A** Multiprotein immunofluorescence demonstrates that Phago.TAM cells in meningiomas express immunosuppressive proteins such as TIM-3 (left, yellow label) and LAIR-1 (right, white label). Representative results are shown; $n = 4$ patient samples were tested from varying WHO grades with similar findings. Scale bar is 100 μM and 20 μM for the inset. **B** Using the same $n = 4$ patient samples, collagens were strongly detected in meningioma stromal regions and exhibited concentric laminar patterns (left). Both tumor and stromal areas harbor cells expressing immune suppressive LAIR-1 - a known receptor for collagens (right). The right image shows an overlapping region from panel A. Scale bar is 200 (left) and 100 (right) μM. **C** Volcano plot showing select chemokines enriched in the dura (left side) and meningioma tissues (right side) based on scRNA-seq of immune clusters shown in Fig. 1A. The annotated chemokines correlate with the elevated myeloid (blue text) and lymphoid (red text) populations in the meningioma and dura, respectively. Differentially expressed genes were calculated through pseudobulk analysis using DESeq2 analysis (two-sided Wald test with Benjamini-Hochberg (BH) procedure for multiple correction) on immune cells grouped to compare between dura and meningioma. **D** UMAP plot of scRNA-seq data from meningiomas and dura reveals three distinctive NK populations associated with resting (green), activated (blue), and exhausted (red) phenotypes. Cell annotation was performed based on DEGs using Seurat FindAllMarkers. **E** Dot plot of selected NK marker genes. Bubble size corresponds to the percent of cells expressing each marker; colors indicate average expression. Activated NK cells exhibited higher levels of granzyme and CD247, which were relatively absent in resting populations. Exhausted NK cells had high expression of both activation and inhibition markers and components of the TGF-β pathway.

---

subgroup, and previous resection, further suggesting that STING is consistently present across the broad spectrum of meningioma subtypes (Figure. S5D). Collectively, these data indicate that the cancer lineage-specific nature of STING chromatin accessibility, alongside differences in promoter methylation levels, is associated with STING expression in meningioma tumor cells.

## STING activation induces direct meningioma cytotoxicity through ROS-induced programmed necrotic cell death

The effects of STING activation were evaluated on ex vivo patient meningiomas ($n = 9$ patients) collected directly from the operating room (Fig. 4A; Figure. S6; Table S1). Treatment of these samples (which included the entire meningioma microenvironment) with the STING agonist 8803 induced progressive cell death over 72 hours, relative to untreated matched controls ($p < 0.01$; Fig. 4B). To clarify if 8803 was causing a direct cytotoxic effect on meningioma neoplastic cells, an additional set of ex vivo human meningioma samples ($n = 3$ patients) were dissociated into single cell suspensions, and the CD45+ immune population was removed using immunoaffinity columns before the initiation of treatment. In the absence of immune cells, 8803 again

induced meningioma cell death at 72 hours ($p < 0.01$), confirming that the effects of this drug were not dependent on co-culture with immune cells (Fig. 4C).

To clarify the underlying mechanism of 8803-induced meningioma cell death, this ex vivo assay was repeated in tandem with a panel of cell death pathway inhibitors, aiming to understand which pathways were essential for full 8803 efficacy. Inhibitors of necroptosis, pyroptosis, and ferroptosis partially reversed 8803-induced meningioma cytotoxicity (Fig. 4D), suggesting that these mechanisms play a role in inducing cell death after STING activation. These pathways, collectively categorized as programmed necrotic cell death[44], can be activated through the induction of ROS[45-47], and STING has been shown to regulate the transcriptional program that controls the generation of ROS[48]. Bulk RNA-sequencing of patient meningioma samples after treatment with 8803 confirmed upregulation of gene signatures associated with autophagy, necroptosis, and pyroptosis (Fig. 4E). RNA-seq also revealed that caspase family genes including CASP1, 4, 7, 8, and 10 were upregulated after treatment with 8803, as well as increased expression of genes that are essential for the necrotic cell death pathway pyroptosis (Fig. 4F).

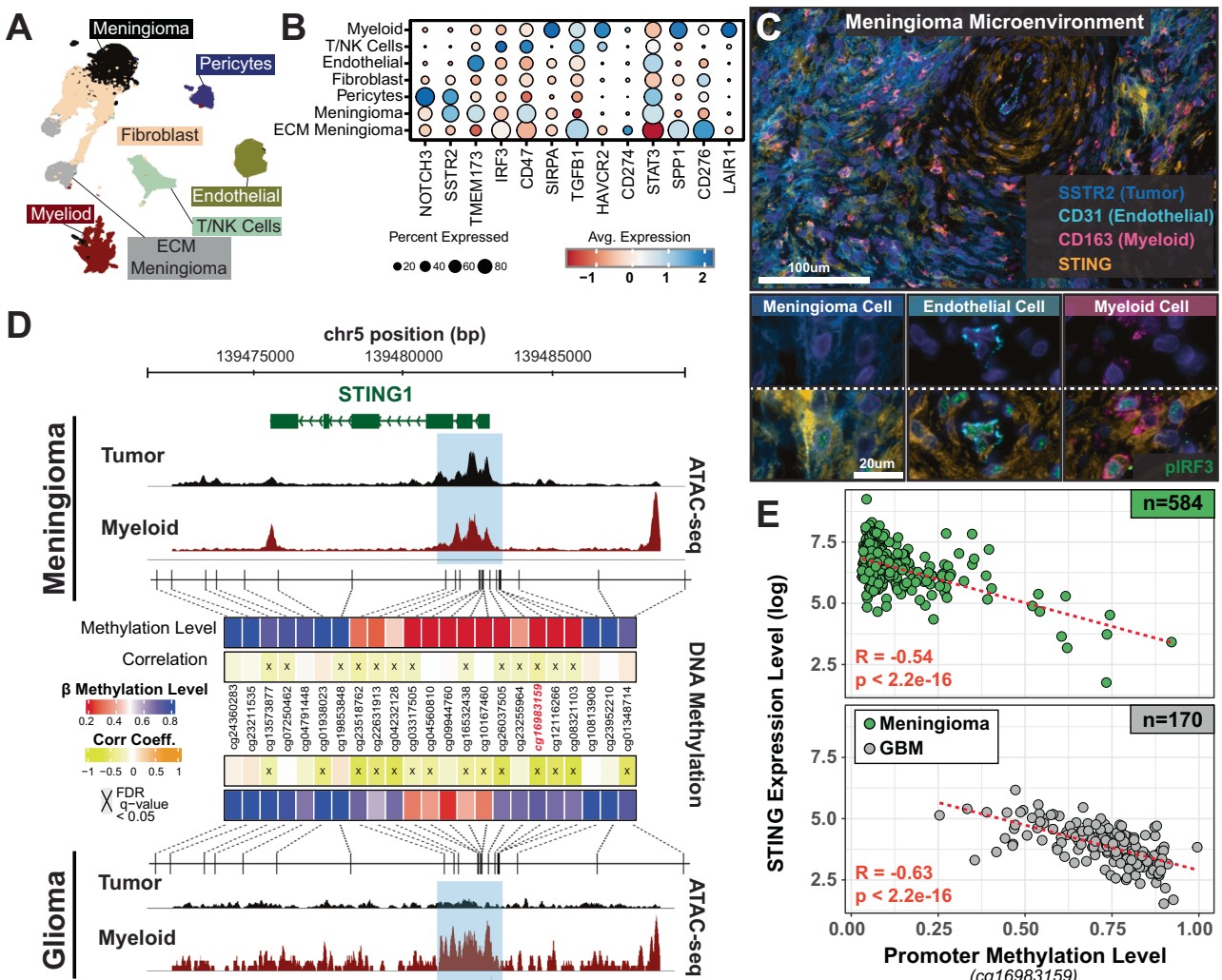

**Fig. 3 | Increased Chromatin accessibility and promoter hypomethylation drives STING expression across meningioma subtypes. A** UMAP cluster analysis of all cells within the meningioma microenvironment distinguishes distinct populations of meningioma, fibroblast, and extracellular matrix (ECM)-producing meningioma cells (n = 22 patients). Cell annotation was based on differentially expressed genes (DEGs) using standard parameters in Seurat FindAllMarkers on all clusters (**B**) Dot-plot demonstrating immune target expression analysis across meningioma cell populations, including the expression of STING (TMEM173) in meningioma, myeloid, and endothelial cells. Bubble size corresponds to the percent of cells expressing each marker; colors indicate average expression. **C** STING (orange) is widely expressed in the meningioma microenvironment, specifically in the neoplastic (SSTR2⁺), endothelial (CD31⁺), and myeloid (CD163⁺) populations. The bottom images also show the nuclear expression of pIRF3 (green) - a marker of downstream STING activation in meningioma, endothelial, and myeloid cells. *n* = 4 patients with similar results. **D** Pseudo-bulk ATAC-seq signal across myeloid and tumor cell populations from meningiomas reveals active chromatin regions near the STING locus (light blue highlight). Similar activation is seen in myeloid populations from glioma (GSE230389)[100]; however, the chromatin near STING is closed in the glioma tumor populations. In the middle, a heatmap depicts the correlation of methylation position with expression in meningioma (top heatmap rows) and glioma (bottom heatmap rows). Significant correlations are indicated by "X" in correlation rows (FDR < 0.05). The strongest methylation probe correlating with STING expression is highlighted (red, bold, italicized): cg16983159. **E** Correlation of cg16983159 methylation level with STING expression among meningioma (*n* = 584 patients) and glioblastoma (*n* = 170 patients) samples, highlighted in green and grey, respectively. Two-sided Pearson correlation was used.

Proteolytic maturation of the pore-forming protein Gasdermin D (GSDMD) by caspase-1 is necessary for triggering pyroptosis[49], and results in oligomerization of the N-terminal of this gene (Fig. 5A). Since caspase-1 was upregulated in meningioma cells treated with 8803, uncleaved and oligomerized GSDMD were quantified by western blot across meningiomas treated with 8803 (*n* = 5 patients) (Fig. 5B, C). There was a small but insignificant difference in total GSDMD in these conditions; however, the ratio of cleaved to total GSDMD was significantly increased upon 8803 treatment (*p* = 0.031). At the transcriptional level, there was a concordant increase in GSDMD expression with 8803 treatment based on bulk RNA-seq data (fold change = 4.64, *p* < 0.001). As previously observed in pyroptosis[50], increases in cleaved GSDMD lead to membrane pore formation, cell death, and leakage of cellular content, including DAMP molecules,

further promoting cytotoxicity and possibly immune activation. Indeed, scanning electron microscopy of 8803 treated meningioma cells demonstrated stepwise pore formation in the plasma membrane over 48 hours that ultimately resulted in cell lysis (Figs. 5D; S7A–D).

Pyroptosis (in addition to necroptosis and ferroptosis) can be activated through the induction of ROS[45–47], and STING has been shown to regulate the transcriptional program that controls the generation of ROS[48]. In meningiomas, 8803 mediated cell death was partially blocked with the ROS inhibitors N-acetyl cysteine (NAC) and GSH ethyl ester (GSHee) (Fig. 5E), indicating ROS production contributes to STING cytotoxicity. As ROS release can be associated with mitochondrial membrane permeability, tunneling electron microscopy was used to examine meningioma cells after administration of 8803. Over 48 hours, there was diffuse blunting of mitochondrial cristae and

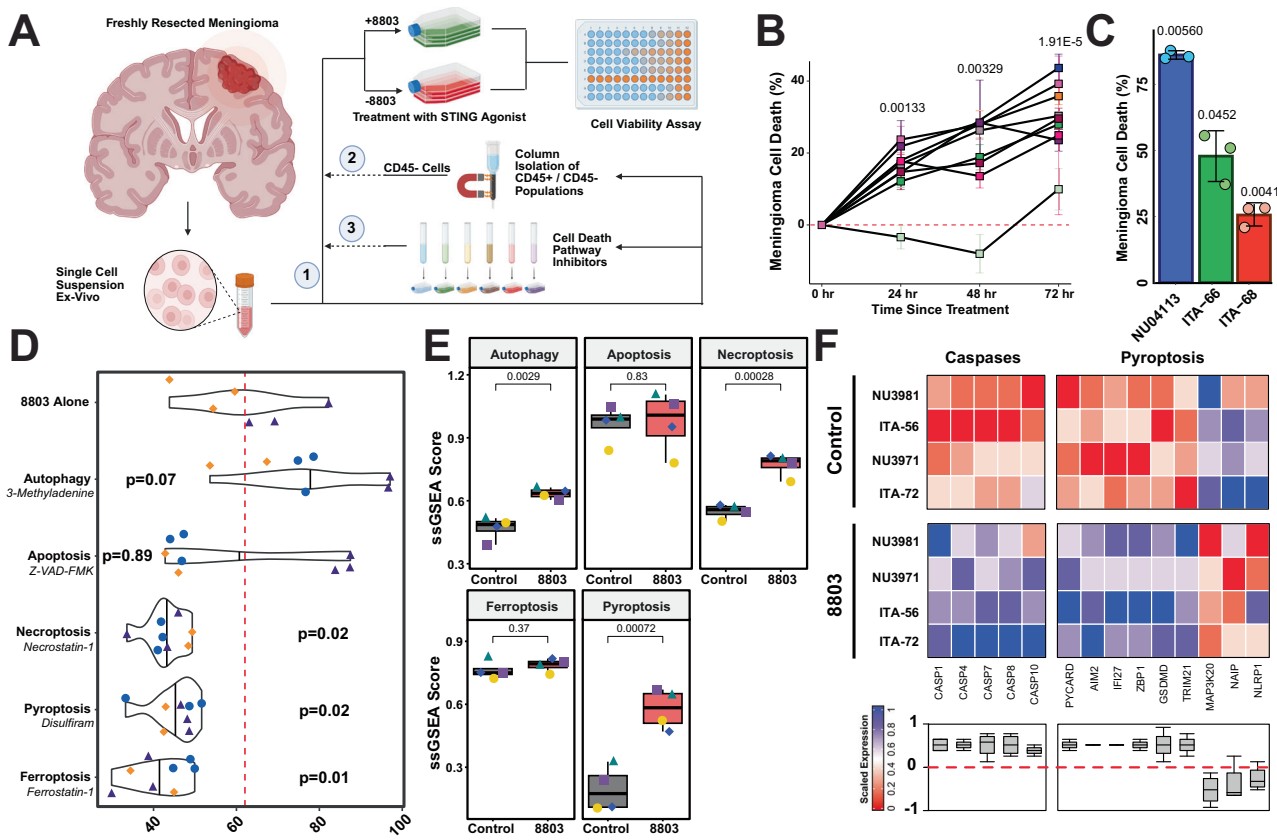

**Fig. 4 | Treatment of ex vivo meningioma with the STING agonist 8803 induces programmed necrotic cell death. A** Schema created with BioRender demonstrating ex vivo experiments of resected patient meningiomas, including (i) treatment of meningioma microenvironment with 8803 (**B**), (ii) removal of CD45+ immune cells before treatment with 8803 (**C**), and (iii) treatment with inhibitors of cell death pathways before treatment with 8803 (**D**). **B** Longitudinal viability assessments of ex vivo meningiomas spanning WHO grades and DNA methylation subtypes treated with 10 μM of the STING agonist 8803 ($n = 9$ patients). Each color and symbol represents a distinct patient (see further details in Fig. S6). The standard deviation is shown as error bars for each time point of each sample, calculated across a median of 10 replicates. $P < 0.01$ from paired t-test among treated and control cells ($P = 1.33 \times 10^{-3}$, $3.29 \times 10^{-3}$, and $1.91 \times 10^{-5}$ for 24 hr, 48 hr, and 72 hr, respectively). **C** Cell viability of ex vivo meningioma specimens ($n = 3$ patients) in which the CD45+ immune cell population is removed before treatment with 10 μM of 8803. The cell death percentage across replicates for each sample was significantly higher than that of untreated controls in the corresponding sample. Each

bar represents the cell killing percentage relative to the untreated control for each patient. Data are presented as mean values +/− SEM. Paired t-test performed between treated and control conditions for NU04113 ($P = 5.60 \times 10^{-3}$), ITA-68 ($P = 4.11 \times 10^{-3}$), and ITA-66 ($P = 4.52 \times 10^{-2}$). **D** Ex vivo meningiomas ($n = 3$ patients) treated with cell death pathway inhibitors before administration of 8803. **E** Single sample Gene Set Enrichment Analysis of bulk RNA-sequencing of patient meningioma samples after treatment with 8803 ($n = 4$ paired patient samples (control vs 8803 treatment)). Each symbol/color represents a different ex vivo meningioma. **F** Genes associated with caspases and pyroptosis that were differentially expressed ($|logFC| > 1$, $p$-value adjusted <0.05) in patient samples ($n = 4$ paired patient samples, control vs 8803 treatment). The range of changes in expression counts per million (CPM) is shown below each gene (log transformed). In (**C**–**E**), data were analyzed using a two-sided T-test. In E and F, the box bounds the interquartile range divided by the median, with the whiskers extending to a maximum of 1.5 times the interquartile range beyond the box.

compromise of the double membrane in these organelles relative to untreated samples (Fig. 5F; Fig. S7E).

## The STING agonist 8803 exerts a therapeutic effect in preclinical meningioma models

We next evaluated the therapeutic efficacy of 8803 in several murine meningioma models. Nude mice were implanted with IOMM-LEE meningioma cells bilaterally in both flanks but were only treated on the right flank with 8803 on day 10 post-tumor implantation (Fig. 6A). Western blot analysis confirmed expression of STING in meningioma cell lines (IOMM-LEE, CH157, MGS1), but as expected, not in glioblastoma cell line controls (0827 and GBM12) (Fig. 6B, C). The treated meningioma showed decreased volume relative to the untreated contralateral tumor and bilaterally untreated mice ($P < 0.01$) (Fig. 6D, E), indicating that there were no systemic anti-cancer therapeutic effects of 8803 with direct intratumoral injection. To evaluate the effects of 8803 at a later time point, the flank experiment was repeated but mice were treated on day 17, rather than day 10. Tumors in the

untreated group continued to grow, while those that received 5 μg of 8803 remained stable ($P < 0.01$) (Fig. 6F).

Next, two orthotopic models were evaluated based on the response to the STING agonist in the flank model. J:NU mice bearing intracranial IOMM-Lee or CH157 meningiomas, which also express STING (Fig. 6B), were treated with the STING agonist 8803. The mice were randomized to be treated with either PBS control or 8803 (5 μg/mouse) injected directly into the tumor starting on day 5 (Fig. 7A). 8803 increased median survival in the orthotopic IOMM-LEE model (8803 median survival (MS): 34 days vs untreated MS: 24.5 days; $P = 0.0125$) (Fig. 7B). Long-term durable survival was observed in the CH157 bearing mice, with four mice treated with 8803 surviving past 150 days (8803 MS: 45 days vs untreated MS: 16 days; $P < 0.0001$) (Fig. 7C). Because Nude mice do not express mature T cells, these data would suggest that the mechanism of action is not fully dependent on T cell-mediated immune responses and that 8803 may be exerting direct cytotoxicity in these in vivo models.

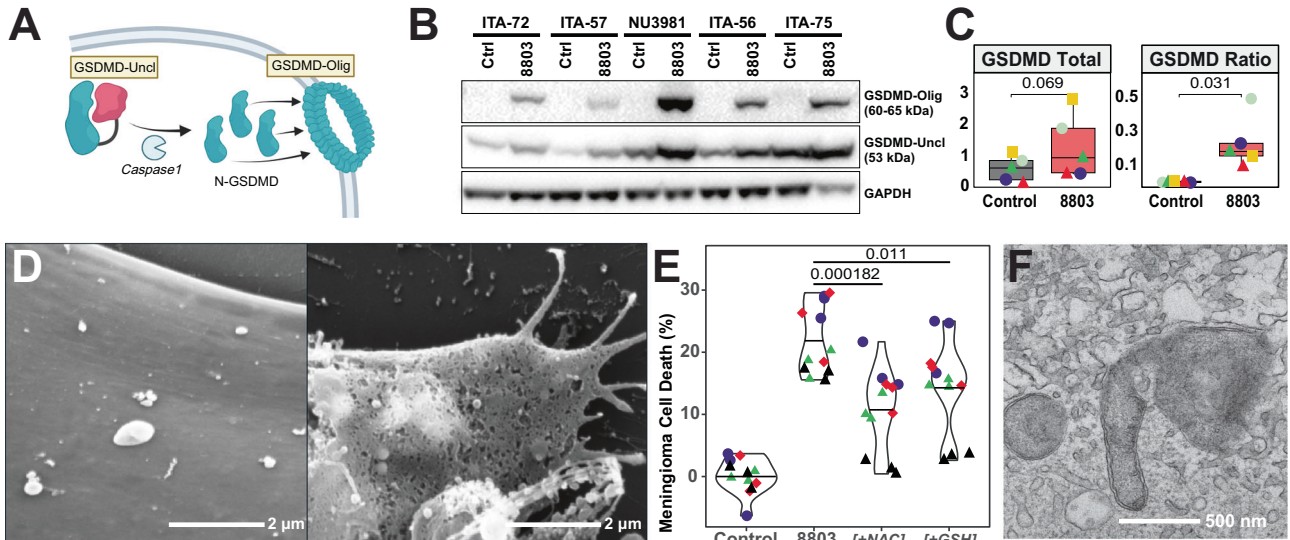

**Fig. 5 | The STING agonist 8803 triggers meningioma pyroptosis through GSDMD induction. A** Schema created with BioRender showing the cleavage of GSDMD via caspase enzymes results in oligomerization and pore formation in the cell membrane, leading to leakage of cellular contents. **B** Western blot analysis of ex vivo meningiomas treated with 8803 ($n = 5$ patients) and then probed for uncleaved or oligodimerized GSDMD with GAPDH as a loading control. The administration of 8803 results in increased oligomerization of GSDMD, consistent with the induction of pyroptosis. **C** Volumetric densitometry data of the Western blot data indicate an increased ratio of oligomerized GSDMD to total GSDMD, normalized to GAPDH ($n = 5$ patients). The box bounds the interquartile range divided by the median, with the whiskers extending to a maximum of 1.5 times the interquartile range beyond the box. **D** Scanning electron microscopy (SEM) of untreated (left) and 8803-treated (right) meningioma cells. Cells treated with 10 µM 8803 for 6 hours display a plasma membrane with irregular pores compromising cellular integrity. For SEM, at least two imaging sessions for each of the time points were performed (control, 6 h, and 48 h) with two replicates per time point. Representative images were selected from what was observed more broadly in the cell sections. **E** Meningioma cell death induced with 8803 is partially blocked with the ROS inhibitors N-acetyl cysteine (NAC) and glutathione monoethyl ester (GSHee). The percentage of cell death at 48 hours is shown for four samples in triplicate (each color/shape represents a different meningioma). A significant decrease in cell killing is observed when 8803 treatment is blocked with NAC ($P = 1.82 \times 10^{-4}$) or GSHee ($P = 0.011$). **F** Representative image of transmission electron microscopy (TEM) of meningioma mitochondria treated for 6 hours with STING-agonist 8803. Internal mitochondrial structures like cristae are collapsed, while a compromised double membrane is evident. For TEM, at least two imaging sessions for each of the time points were performed (control, 6 h, and 48 h) with two replicates per time point. Scale bar: 500 nm.

## STING activation triggers local inflammation and reshapes the meningioma microenvironment

The therapeutic activity of 8803 was next evaluated in an immuno-competent, syngeneic, murine mouse model[51], via orthotopic implantation of MGS1 cells in FVB/N background mice. A dose of 5 µg of 8803 prolonged survival in treated mice (8803 MS: undefined vs untreated MS: 39 days; $P < 0.001$) (Fig. 8A). On day 48, the experiment was terminated secondary to meningioma tumor growth outside of the skull in some animals. At this point, the extent of the intracranial burden was analyzed. 8803-treated mice had minimal tumor growth intracranially, whereas all control mice had extensive growth (Fig. 8B), including within the brain (Fig. 8C). Additionally, we observed inflammatory cells present in mice treated with 8803, while those treated with PBS alone exhibited possible brain invasion. During the therapeutic window of treatment with 8803 (on day 28), a parallel cohort of mice was terminated and perfused, and the meningioma and the adjacent brain were dissected for single-cell analysis (2 samples per condition; total of 81,170 cells). Within the CD45+ immune compartment, 8803 induced a marked expansion of macrophages, NK, and lymphoid immune cell populations (Figs. 8D, E, S8A–C, and Table S2). Adjacent microglia shifted from a homeostatic to an early activation and interferon-stimulated gene (ISG) signature state. Across immune lineages, 8803 increased the presence of immune effector responses within the meningioma (Fig. 8F). These findings are consistent with the known functions of STING agonists, including activation of macrophages, NK cells, and cytotoxic T cells.

Unbiased gene expression profiling demonstrated 8803 markedly upregulated IFN, perforin, and MMP but down-regulated immune suppressive Havcr2 (i.e., TIM3) and Lair1 (Fig. 9A). When the murine meningioma tumor cells treated with 8803 were analyzed, after alignment with human gene signatures (Fig. S8A), gene expression profiling was notable for the induction of caspases, reactive oxygen species, STING, and GSMD, but a reduction of collagen (Fig. 9B, C). These data were consistent with human meningioma specimens treated with 8803 (Figs. 4F, S9).

## Meningioma lysis triggers macrophage activation and ECM breakdown through the TLR and STING pathways

The immune modulatory effects of STING agonists have been extensively described; however, the synergistic effects of tumor lysis in this setting are not well understood. Since 8803 mediates direct tumor cytotoxicity in meningiomas, DAMP molecules may trigger inflammatory responses through pattern recognition receptors such as toll-like receptors (TLR). To evaluate the effects of meningioma DAMPs on macrophage activation, we devised a strategy whereby isolated macrophages from FvB WT, MyD88-/-, or STING-deficient Tmem173gt background mice were treated with control media or lysed MGS1 cells. This approach circumvented the direct effects of 8803 on macrophages (meningioma cells were lysed using a mechanical technique), while the use of wild-type, TLR-deficient (MyD88-/-) and STING-deficient (Tmem173gt) macrophages permitted independent assessment of these components in macrophage activation. Across a Luminex panel of cytokines and chemokines, VEGF, and MMP-3 demonstrated marked physiologically relevant and consistent increases in macrophages when exposed to the MGS1 lysate (Figs. 9D; S10, S11). MMP-3 activates MMP-9, which degrades collagen[52–57] that forms a substantial component of the meningioma stroma (Fig. 2B). MMP-3 relies on the TLR pathway for activation in the setting of DAMPs, since production was markedly attenuated in the MyD88-/-deficient macrophages (Fig. S10). As such, when a STING

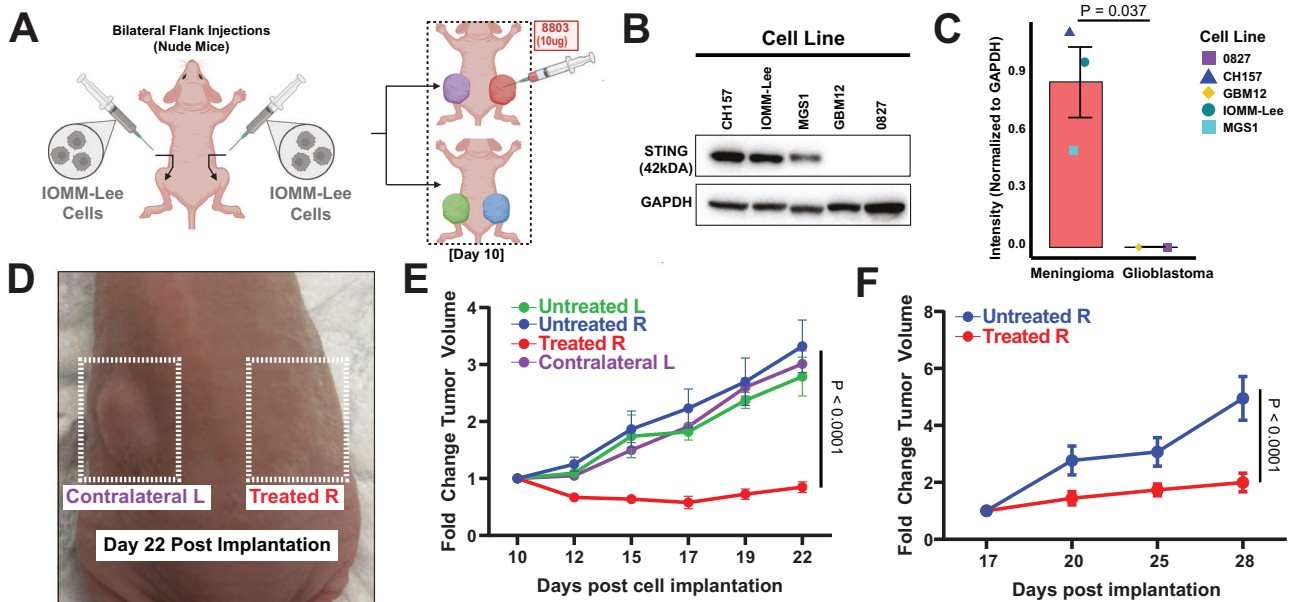

**Fig. 6 | 8803 exerts a therapeutic effect in a preclinical meningioma model.**
**A** Efficacy of 8803 on the IOMM-Lee cell line was evaluated in immunocompromised nude mice. Schema created with BioRender. On day 10, one cohort of mice received unilateral intratumoral injection of 8803 (red), while the contralateral side received a PBS injection (purple). In a second cohort used as controls to evaluate any systemic effects of 8803 in the first cohort, both flank tumors were treated with PBS sham (green and blue). **B** Western blot analysis of meningioma (CH157, IOMM-LEE, MGS1) and glioblastoma cells (GBM12 and 0827) probed for expression of STING. **C** Densitometry results from Western blot confirming STING expression in meningioma cell lines ($n = 3$ cell lines: CH157, IOMM-Lee, MGS1) and glioblastoma cell lines ($n = 2$ cell lines: 0827, GBM12, 0827) relative to GAPDH ($p = 0.037$). Data are presented as mean values +/− SEM and analyzed using a two-sided T-test.
**D** Representative photo of a mouse on post-implant day 22 showing minimal tumor burden on the treated right side (red) but an evident tumor on the

contralateral PBS-treated left side (purple) ($n = 5$ mice). **E** Graph of tumor volumes of bilaterally implanted IOMM-Lee cells. Mice ($n = 5$ mice/group) were treated with $5\,\mu g$ of 8803 on day 10 (red). Tumors in all other groups (green, blue, purple) continued to grow, but the 8803-treated tumors remained stable ($P < 0.0001$ for time and $P = 0.000022$ for treatment). Data are presented as mean values +/− SEM and analyzed using two-way ANOVA with Tukey's multiple comparisons. **F** Graph demonstrating the tumor volumes of the flank model of unilateral implanted IOMM-Lee cells in which only one side is initially treated at a later time point, starting on day 17. Mice ($n = 7$–$10$ mice/group) were treated with $5\,\mu g$ of 8803. The tumor in the untreated mouse continued to grow, but the 8803-treated tumor remained stable ($P < 0.0001$ for time and treatment). Experiment terminated on Day 28 due to the control tumors ulcerating and reaching a humane endpoint. Data are presented as mean values +/− SEM and analyzed using two-way ANOVA with Tukey's multiple comparisons.

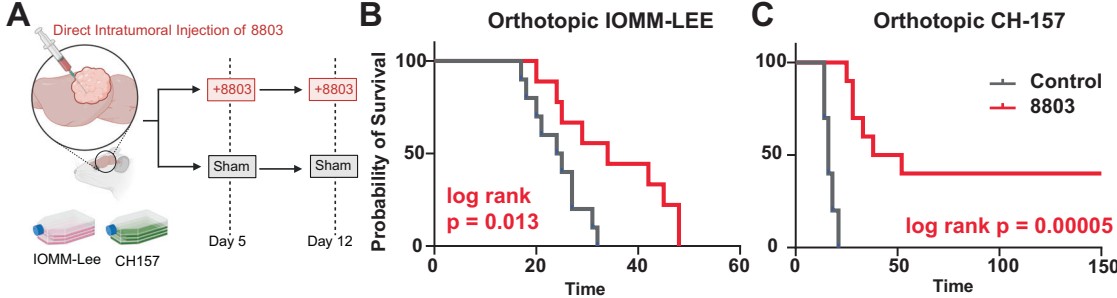

**Fig. 7 | 8803 exerts a therapeutic effect across orthotopic preclinical meningioma models.** **A** Schematic of orthotopic treatment experiments in immunocompromised mice created with BioRender. **B** Survival of nude mice orthotopically implanted with IOMM-Lee using Kaplan-Meier analysis. Controls: $n = 10$ mice (median survival [MS]: 24.5 days), 8803: $n = 9$ mice (MS: 34 days). Statistics (log-

rank test): control versus 8803, $P = 0.0125$. **C** Survival of nude mice orthotopically implanted with CH157-MN using Kaplan-Meier analysis. The remaining mice were terminated at day 150. Controls: $n = 10$ mice (MS: 16 days), 8803: $n = 10$ mice (MS: 45 days). Statistics (log-rank test): control versus 8803, $P = 0.00005$.

agonist such as 8803 induces tumor cell death, immune responses are potentiated through the TLR pathway.

To ascertain if the TLR and STING pathways are working in synergy through DAMP-related and 8803-mediated processes in macrophages, an additional Luminex assay was performed using wild-type (FVB) macrophages treated with media (control), lysed MGS1 meningioma cells, 8803, or the combination (Fig. S11). Interaction effects in linear regression models revealed that MMP-3, VEGF, and TWEAK were primarily driven by exposure to MGS1 lysate, while CCL2

and CXCL10 were due to the 8803 treatment. Notably, TNF-α, IL-6, and CXCL9 exhibited manifold effects from the combination of 8803 and MGS1 lysate, suggesting synergy between these exposures. These results indicate that the treatment of meningiomas with 8803 induces inflammatory cytokines via combined mechanisms, including direct action of this agonist and release of DAMPs from necrotic meningioma cells to activate macrophages.

As described previously[58], human meningiomas exert significant mass effect through the presence of collagen, including COL6A

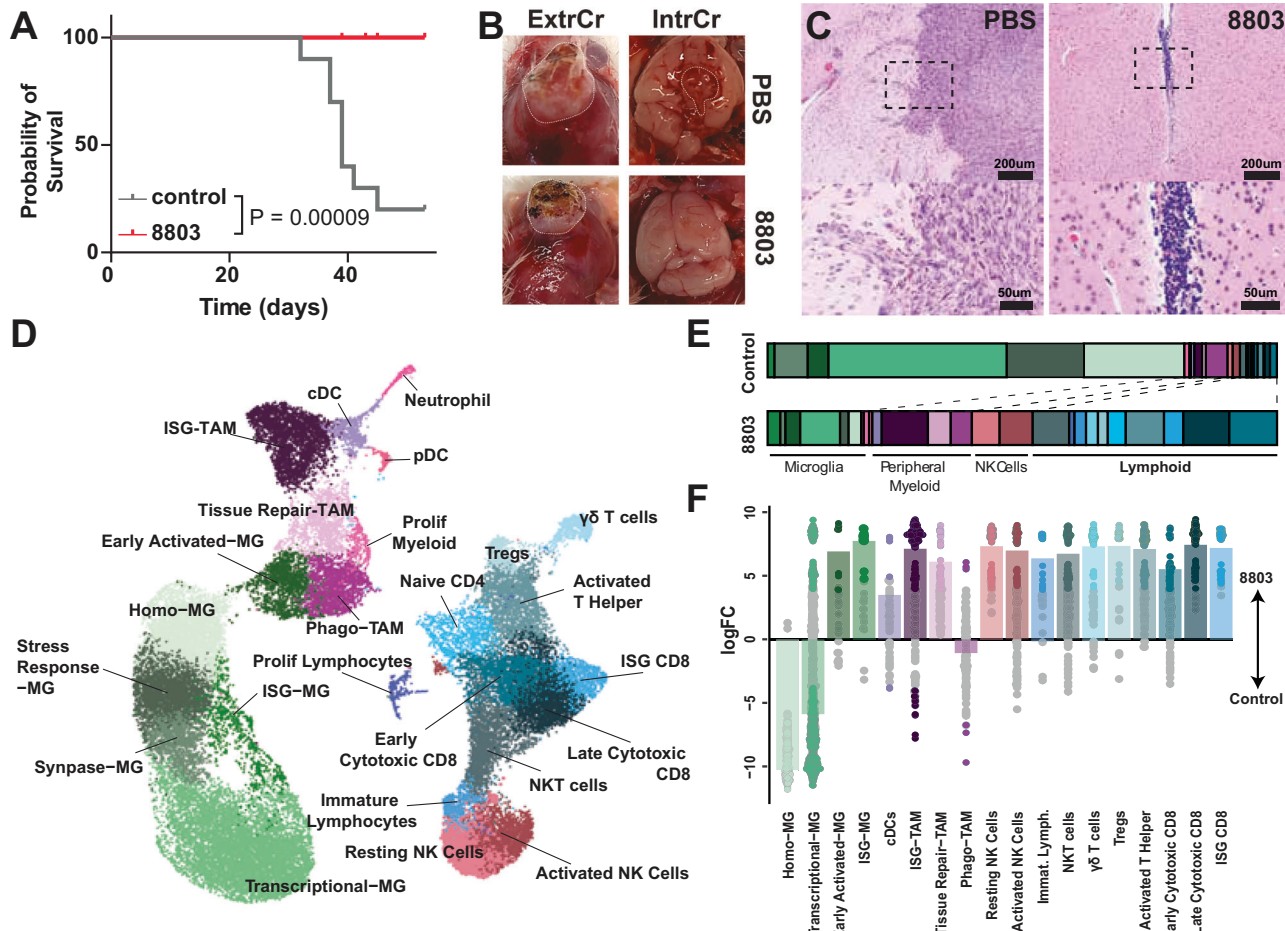

**Fig. 8 | Orthotopic STING activation transforms the meningioma microenvironment. A** Survival of immunocompetent FVB/N mice orthotopically implanted with MGS1 using Kaplan-Meier analysis. Controls: $n = 10$ mice (median survival [MS]: 39 days), 8803: $n = 9$ mice (MS: undefined days). Statistics (log-rank test): control versus 8803, $P = 0.0009$. **B** Representative images of extracranial (ExtrCr) and intracranial (IntrCr) tumor burden in control and 8803 treated MGS1-bearing mice. Though both conditions exhibited ExtrCr growth of tumors away from the region of injection, only PBS-treated mice had IntrCr tumor burden on post-mortem analysis. **C** Representative H&E image of control and 8803-treated MGS1-bearing mice demonstrating the presence and absence of intracranial meningiomas. Mice treated with 8803 demonstrated immune infiltration, while PBS-treated mice exhibited brain invasion, as shown in the bottom insets. $n = 3$ mice for PBS control and $n = 3$ mice for 8803 treatment. **D** UMAP plot of scRNA-seq data of 42,150 immune cells (CD45+). $n = 6$ mice per condition (untreated and 8803).

Three mice were pooled for each sequencing run. Cell annotation based on differentially expressed genes (DEGs) using standard parameters in Seurat FindAll-Markers on all clusters (Table S2). Meningioma tumor cells were additionally identified using FindTransferAnchors using human scRNA (analyzed in Fig. 1) as a reference database. ISG: Interferon-Stimulated Genes; TAM: tumor-associated macrophages (**E**) Stacked bar graph and (**F**) Strip plot demonstrating that 8803 increases cytotoxic immune populations. Differential abundance testing was conducted on batch corrected sample level clusters using MiloR standard workflow, including: buildgraph, makeNhoods, countcells, calcNhoodDistance, and testN-hoods. Visualization was performed using plotDAbeeswarm. Neighborhoods are colored if *p*-value < 0.05; otherwise shown in grey. Immune cell populations with fewer than 5 significant neighborhoods were removed from the strip plot. A two-sided negative binomial generalized linear model with spatial FDR correction was used.

(Figs. 2B; S4). Although 8803 may induce tumor cell death, the persistence of collagen could be a confounder for symptomatic treatment responses. As transcriptional results from human and mouse meningiomas suggested decreased collagen production and increased MMP production in response to 8803, and production of MMP-3 was also seen in macrophages via Luminex, we performed multiplex immunofluorescent analysis to interrogate collagen in murine MGS1 meningiomas after orthotopic treatment. Control-treated intracranial tumors demonstrated marked COL6A1/A2 expression throughout the tumor (Fig. 9E, left), similar to human meningiomas. In contrast, 8803-treated meningiomas had virtually no COL6A1/A2 expression (Fig. 9E, right) among all mice investigated ($n = 3$ mice for each condition; Fig. S12). These data indicate that standard MRI imaging could be used as a therapeutic endpoint in a clinical trial of 8803 for meningiomas. Collectively, our results demonstrate that STING agonist 8803 exhibits complementary therapeutic mechanisms in meningiomas that include

direct cytotoxicity, immune activation, and degradation of the tumor stroma (Fig. 9F).

## Discussion

Previous studies have leveraged bulk tissue datasets or flow cytometry to deconvolute cellular heterogeneity in meningiomas, including immune populations. Immunosuppressive myeloid cells have been recognized as the dominant constituent[14,59–61], with relatively sparse and exhausted lymphoid effector populations such as cytotoxic T cells[34,35,62]. Indeed, our analyses indicate robust enrichment of HSP- and phago.TAM populations that express TIM3 and LAIR-1, confirmed using spatial multi-protein immunofluorescence across diverse types of meningiomas. Multiple forms of collagens were also abundantly present, often in stromal compartments, that tended to associate with LAIR-1 and thereby increase immunosuppression. The preferential enrichment of macrophages within

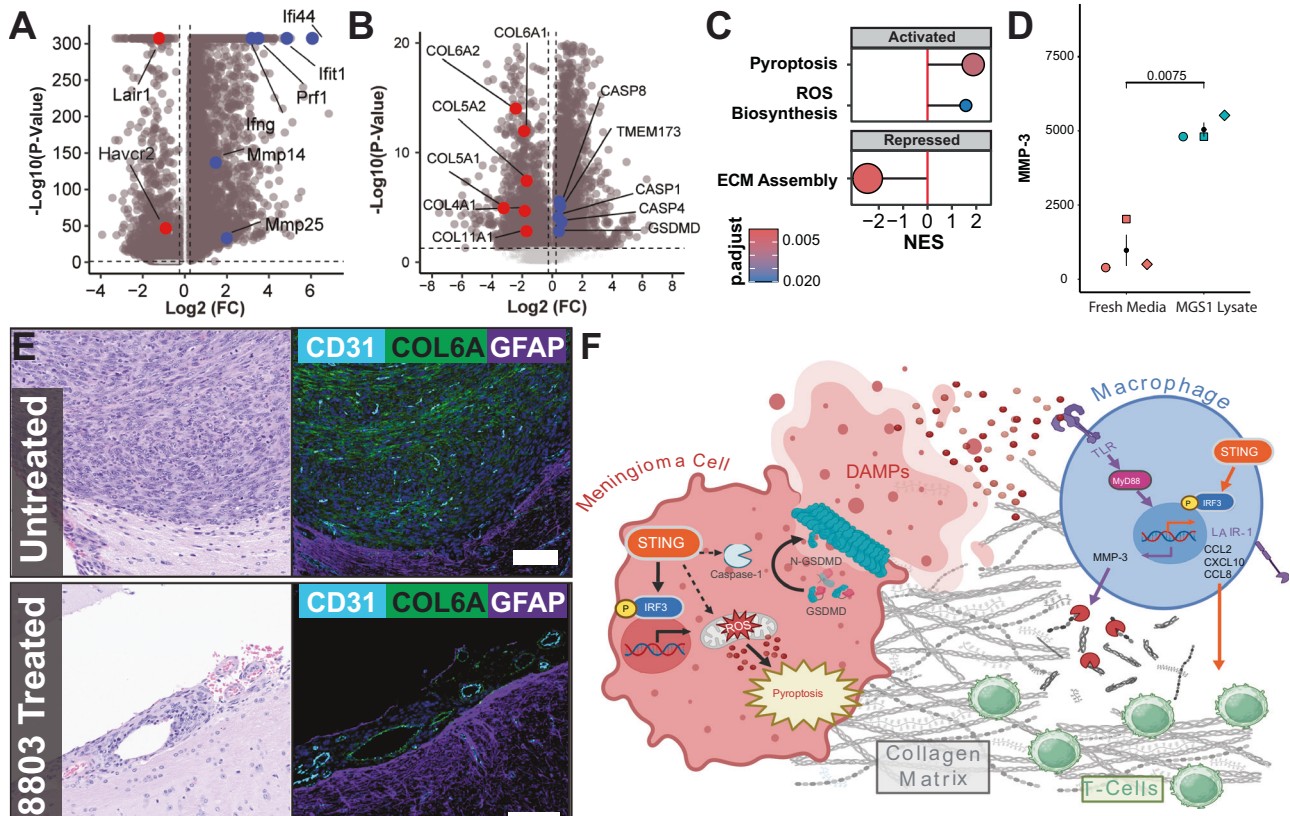

**Fig. 9 | Meningioma lysis triggers macrophage activation and extracellular matric breakdown through TLR and STING pathways. A** Volcano plot of DEGs from immune cells showing that cytotoxic genes such as Prf1 and Ifng, and matrix metalloproteinases are highly upregulated, whereas the immune suppressive molecules like TIM3 and LAIR are decreased following 8803 treatment. Differentially expressed genes were calculated using Seurat FindMarkers using the two-sided Wilcoxon Rank Test between 8803 and control from immune cells. Data was also analyzed using DESeq2 and presented in Fig. S8D. **B** Volcano plot of DEGs from meningioma cells showing a decrease in several notable collagens and an increase in STING (Tmem173), GSDMD, and cell death caspases. Differentially expressed genes were calculated using Seurat FindMarkers using the two-sided Wilcoxon Rank Test between 8803 vs control from meningioma tumor cells. Data was also analyzed using DESeq2 and presented in Fig. S8E. **C** Dot plot of GSEA results showing increased pyroptosis and ROS generation ('Reactive oxygen species bio-synthetic process') and decreased ECM assembly in meningioma cells following 8803 treatment. NES: Normalized Enrichment Score. Analyzed using weighted

Kolmogorov-Smirnov test with FDR correction (**D**) Luminex assay reveals elevation in MMP-3 (a protease that degrades extracellular matrix) in response to treatment of wild-type (FvB) macrophages with MGS1 lysate (n = 3 biological replicates). Data are presented as mean values +/− SEM and analyzed using a two-sided T-test. See additional assays and conditions in Fig. S11-S12. **E** Treatment of MGS1 tumors with 8803 (n = 3 mice) resulted in a decrease in COL6A (right) compared to control (left) (n = 3 mice). COL6A is a key component of human meningioma stromal mass. Scale bars represent 100 µm. Adjacent H&E slides are shown for each tumor slice. These panels are also shown in Fig. S12 at different magnifications, along with additional examples. **F** Graphical summary created with BioRender demonstrating that the administration of the STING agonist 8803 triggers mitochondrial ROS-mediated programmed necrotic death pathways (pyroptosis, necroptosis, ferroptosis) in meningioma cells, resulting in DAMP release via GSDMD pore formation that activates latent macrophages to produce collagen-degrading enzymes and attract T cells through chemokine elaboration.

---

meningiomas suggested innate immune modulatory strategies as an appropriate therapeutic starting point in these tumors, especially given the frequency of the phago. TAM population and the lack of effector T cells. In this context, we identified widespread expression of STING in the meningioma microenvironment, a key inducer of type I interferon with potent capacity to trigger local inflammation. The presence of STING in both immune and meningioma cells suggested a unique therapeutic opportunity to induce both direct tumor and inflammatory immune responses, which distinguishes meningiomas from many other solid tumors, such as glioblastoma. This also provided a chance to study the cytotoxic capacity of the cGAS-STING pathway in neoplastic cells, thereby elucidating downstream mechanisms that underlie clinical utility.

The first clinical trial of a STING agonist for solid cancer used the ADU-S100 compound, which was injected directly into meta-static tumors. Although well tolerated and with the injected lesions showing either stability or regression, there were only a few patients with overall partial survival responses[63]. Other STING agonists such as

BI 1703880[64], MK-2118[65], E7766[66], and IMSA101[67] have also been developed and evaluated in Phase I clinical trials, but the signals of response have remained modest. In contrast to these previously investigated indications, meningioma is usually a local, solitary disease. Because meningiomas generally demonstrate chromosomal stability and low basal levels of STING activation, the STING pathway may be primed to respond to an agonist, thereby yielding tumor control. Most likely these responses are based on a variety of factors including epigenetic regulation of the STING promoter, STING isoforms and mutations, immune composition of the TME, and other regulatory factors that are now being elucidated. To investigate the therapeutic potential of this pathway in meningiomas, we used the agonist 8803, a 2',3'-thiophosphate CDN analog, which shows superior and robust activation of the STING pathway based on benchmarking to ADU-S100[18,68]. While therapeutic resistance to STING agonists can be acquired through mutations and alterations of STING isoforms that render them insensitive to stimulation[69], 8803 can still generate immune activation in this scenario. As such, the use of 8803 in

meningiomas represents a unique therapeutic opportunity that holds promise relative to previous clinical indications.

Though a previous study suggested that long-term STING induction may promote tumor growth[70], acute administration of 8803 in an immunocompetent setting is expected to be safe and efficacious. Indeed, STING agonists have not shown tumor-promoting effects across a variety of preclinical models, nor has this been reported in clinical trials. Pharmacological modulation using low pulsed dosing with STING agonists is critical for boosting T cell and antigen-presenting responses without inducing cell cycle-limiting and immunosuppressive feedback due to excess IFN signaling and high levels of neutrophil activation. Nonetheless, radiographic imaging of meningiomas in clinical studies using 8803 should more frequently assess for the possibility of tumor-promoting effects, although this would be unlikely given the intermittent nature of therapeutic delivery.

Investigation of STING agonists have mostly focused on their immune-mediated anti-cancer activities. In liquid cancers such as leukemia, activation of the STING pathway can directly trigger neoplastic cell apoptosis[24,25], but this mechanism of cell death does not induce inflammatory responses. STING activation has also been associated with other forms of cell death, such as necroptosis[26,27], autophagy (to downregulate innate immune responses)[28], lysosomal cell death (in myeloid populations)[71], and ferroptosis (during kidney ischemia)[72] that depend on context. Using freshly resected patient meningiomas and animal models, our data demonstrate that STING activation can trigger direct tumor cytotoxicity through the induction of inflammatory cell death pathways, when the STING promoter has not been epigenetically silenced. This conclusion was supported by co-administration of inhibitors for necroptosis, ferroptosis, and pyroptosis in ex vivo assays, transcriptional analysis of 8803-treated meningiomas, and Gasdermin D membrane pore formation. Therapeutic efficacy of 8803 was maintained when CD45+ immune cells were removed before treatment, and when meningiomas were exposed to 8803 in immune-deficient mice, indicating 8803 has direct tumor effects. 8803-induced STING activation triggers Gasdermin D, which damages both the membranes of the cell and mitochondria. This permeabilization leads to reduced mitochondrial membrane potential, increased ROS, and the release of mitochondrial DNA. The mitochondrial DNA leakage into the cytoplasm of the cells likely triggers additional STING activation through cGAS, thereby amplifying the response. Mitochondrial damage and the production of ROS are essential for mediating multiple types of cell death, including necroptosis, pyroptosis, and ferroptosis[73]. Programmed necrotic pathways converge on the generation of ROS associated with mitochondrial disruption[74–77], and we observed that inhibition of ROS production limited the therapeutic efficacy of 8803 in meningiomas. Pyroptosis, a type of programmed necrotic cell death, is mediated by caspase-1[78], which cleaves GSDMD into fragments that oligomerize to form plasma membrane pores[79] and result in leakage of intracellular contents[80]. Indeed, we observed marked upregulation of oligomerized GSDMD after treatment of meningioma cells with 8803, and the knockdown of STING has previously been shown to reduce caspase-1-GSDMD functions[81]. Consistent with these effects, we observed stepwise accumulation of membrane pores after 8803 administration on electron microscopy, ultimately resulting in direct cell destruction. Notably, others have shown that GSDMD is an executor of multiple cell death pathways secondary to mitochondrial dysfunction that initiate immune responses through cell death modality switching[82]. Electron microscopy of 8803-treated meningioma cells revealed blunting of mitochondrial cristae and loss of the double membrane, and our pharmacologic inhibitor and RNA-seq experiments demonstrated that these other cell death pathways also contribute to STING cytotoxicity.

In all these forms of programmed necrosis (pyroptosis, ferroptosis, necroptosis), the leakage of DAMP molecules into the meningioma microenvironment can trigger an inflammatory response that is not observed in other cell death pathways, such as apoptosis. We hypothesized that this might have a synergistic effect with the well-established direct actions of STING activation in immune cells, thus strengthening the therapeutic efficacy of 8803 in meningiomas. Using macrophages isolated from wild-type (FVB) and TLR-deficient and STING-deficient mice, we found that meningioma DAMPs triggered key inflammatory cytokines that depended on TLR signaling. Notably, some cytokines like MMP were specifically induced by DAMPs, while others were more strongly activated by 8803 directly, thus demonstrating the involvement of both the TLR and STING pathways in triggering inflammation. Changes in immune infiltration and activation were confirmed using scRNA-seq of murine meningiomas after 8803 treatment, including down-regulation of key immune-suppressive molecules such as LAIR-1 and HAVCR2. A limitation of this data is the filtering of dead/dying cells with high mitochondrial RNA (standard for scRNA analysis), which may have obscured the detection of specific cell death pathways in this model.

A notable finding of this experiment was changes in genes that regulate the extracellular matrix, such as downregulation of collagen production in meningioma cells and upregulation of MMP genes in macrophages. This latter discovery was confirmed using the Luminex assay under carefully controlled conditions, including TLR and STING KO macrophages. The immunocompetent MGS1 models only grow in the FVB/N background. As such, implantation of MGS1 into backgrounds that have genetically modified alterations in collagen or metalloproteinases will not support tumor growth. To overcome this limitation, we used macrophages from TLR and STING KO backgrounds that were treated with 8803 and/or meningioma lysate. The data indicates that collagen-degrading MMP-3 is induced by the MGS1 lysate, which is abrogated in MyD88 KO macrophages, suggesting that this effect requires the TLR pathway. Treatment of macrophages with 8803 does not trigger MMP-3, indicating meningioma cell death facilitates collagen degradation. Unlike hypercellular tumors such as glioblastoma, the tumor bulk in meningiomas is comprised of collagen and other ECM proteins. Consequently, the reduction of ECM components may reduce symptomatic mass effect. To our knowledge, the role of STING in ECM degradation has not been appreciated in solid cancers and suggests a mechanism by which 8803 may be clinically efficacious. In meningiomas, changes in tumor volume can be utilized as a radiographic endpoint in future clinical trials of STING agonists alongside quality-of-life metrics such as neurological symptom alleviation, thus circumventing traditional metrics that require decades of follow-up. In conclusion, we identify and characterize STING as a potent therapeutic approach in meningiomas mediated through direct tumor cytotoxicity, immune activation, and modulation of tumor stroma. Meningiomas may be an appropriate disease indication for understanding the role and mechanisms of cGAS/STING in tumor biology since they exhibit numerous intrinsic features that render them vulnerable to therapeutic targeting of this pathway. The translation of this approach into human meningioma patients in future clinical trials will offer an approach for this common type of brain tumor.

## Methods

### Ethics Statement

The Northwestern University Institutional Animal Care and Use Committee (IACUC) approved all animal protocols, and all animal care followed institutional guidelines. For flank models, the tumor was used with a Hamilton syringe, with treatment initiated as designated in the schemas. Tumors were measured with calipers in two directions. Flank tumor volumes were calculated using $(LxW^2)/2$. The total size of 3.0 cm in diameter of all tumors combined in mice was not exceeded per IACUC guidelines. Animals bearing tumors that interfered with normal functions, such as eating, ambulating, and eliminating waste, were considered humane endpoints. Rodents with functional

impairments were immediately euthanized. Ulcerated tumors were considered an automatic endpoint. For orthotopic models, animals were when they became moribund which was defined as >20% weight loss, neurologic symptoms, or evidence of pain/distress as per Northwestern IACUC.

## Study approval and cohort collection

These research studies comply with all relevant ethical regulations. Collection and analysis of samples for this study were conducted under protocols STU00211542 and STU00214485, approved by the Northwestern University Institutional Review Board (IRB). Samples that underwent molecular profiling and/or sequential multiplex were collected by the Nervous System Tumor Bank (NSTB) under an additional IRB-approved protocol. Animal studies were approved by the Northwestern Institutional Animal Care and Use Committee (IACUC). Due to specimen availability, not all assays could be run across all ex vivo samples. Human meningioma samples were selected based on the availability of tissue for characterization, and both male and female subjects were included. For all mice studies, both sexes were included.

## Joint single-nucleus RNA-seq and ATAC-seq profiling of meningiomas

Nuclei were isolated from previously frozen meningioma specimens ($n = 4$) using the 10x Chromium Nuclei Isolation Kit with a RNase Inhibitor (PN-1000494) according to the manufacturer's protocol. Samples were selected for profiling based on RNA integrity number (RIN) values > 7 on bulk RNA isolation. After isolation, the nuclei were analyzed using the Nexcelom Cellometer Auto2000 with the AOPI fluorescent staining method, followed by tagging of open chromatin regions with Tn5 transposase. Each sample was then processed on an independent well of the Chromium Next GEM Chip J for GEM generation and molecule barcoding. Both RNA and ATAC-seq libraries were constructed using Chromium Next GEM Single Cell Multiome ATAC + Gene expression kit (10X Genomics, PN-1000281). After library construction, RNA-seq reads were captured by paired barcodes with unique molecular identifiers and ATAC-seq reads labeled with 10x Genomics cell barcodes. Sequencing was performed on the Illumina platform Novaseq X Plus at the Northwestern University NUSeq sequencing facility core, with a targeted sequencing depth of 20,000 and 25,000 reads per sample acquired for RNA and ATAC profiling, respectively. A total of 10,000 nuclei were targeted for each sample, with 25,000 scATAC-seq and 20,000 scRNA-seq reads in each nucleus.

## Analysis of single-cell RNA-seq and ATAC-seq data

10x multiome data was demultiplexed using the Cellranger-ARC pipeline with the mkfastq command used to perform raw base calling of both RNA and ATAC reads in each nucleus. Subsequent alignment, quantification, and peak calling were performed using the count command on each fastq file. The data were loaded in R using the Seurat software package. Cells were filtered based on the following criteria: <2.5% mitochondrial DNA, RNA count range of 500 to 25000, and ATAC count range of 1000 to 25,000. RNA assays for cells were subject to SCTransform for normalization/feature selection and PCA for dimensionality reduction (Choudhary & Satija, 2022). ATAC assays for cells were subject to TF-IDF for normalization, FindTopFeatures for feature selection, and RunSVD for dimensionality reduction. Reduced dimensions for ATAC and RNA data were then normalized for batch effects using Harmony[83]. The UMAP algorithm was used to visualize both RNA and ATAC data separately. The Weighted Nearest Neighbors algorithm was used to integrate RNA and ATAC assays for multimodal clustering. Cell clusters were annotated using the ScType R package with a combination of meningioma cell type markers from previous work (Choudhury, 2022) and canonical nontumor lineages. Separately, these analyses were performed on glioma samples from publicly available data (GSE230389).

Additional single-cell RNA-seq data were extracted from publicly available repositories[2,3] and processed according to prior descriptions[2]. Briefly, the Seurat R Package using the scRNA-seq Seurat10x genomic workflow was used for all analyses unless noted otherwise[84]. Cells were filtered using a percent mitochondrial DNA threshold of 20% and a UMI range of 500 to 10,000. Cells were then subject to Log Normalize, Scale Data, and PCA functions. The FindClusters and FindMarkers functions were utilized for clustering and marker identification, and non-linear dimensional reduction techniques were applied to visual data in UMAP plot format. The Harmony algorithm was used to regress batch effects[83]. Tumor cells were identified using CONICSmat[85] to determine copy-number loss of chromosome 22q using repetitions = 100 and postProb = 0.75. Cells with postProb <0.15 were considered tumor and > 08.5 as normal. Clusters with more than 80% normal cells were considered non-meningioma clusters. The fibroblast cluster had an intermediate proportion of cells with a loss of 22q and is likely a mixture of both normal and tumor cells. Non-tumor cell clusters were annotated using three methods to produce robust cell assignments: 1) comparison against known cell markers; 2) examination of DEGs against the Human Protein Atlas; and 3) ScType R package, an automated cell assignment algorithm. MiloR package was utilized for differential abundance testing using standard parameters and protocol[86]. QC and harmonization data are shown in Fig. S1.

## Bulk DNA-methylation profiling and analysis

DNA was extracted from 508 meningiomas resected at Northwestern Memorial Hospital, followed by profiling on the Illumina Infinium platform (MethylationEPIC BeadChip v1 or v2). IDAT format files for each sample were imported into R (v 4.3.1) using the SeSAMe R package (Bioconductor v3.18)[87]. Probes were normalized within samples using the standard SeSAMe preprocessing pipeline, including normal-exponential out-of-band background correction, nonlinear dye bias correction, and out-of-band array hybridization masking to produce methylated and unmethylated signals. Probes located on the X and Y chromosomes were filtered out. Samples were merged into a single dataset by selecting for probes present on all array types (EPIC v1 or EPIC v2), totaling 705,536 probes. Unmethylated and methylated signals were then normalized for array type by applying linear modeling to the log2-transformed intensity values via the limma R package[88]. β-values were calculated from normalized methylated and unmethylated signals using an offset of 100 as recommended by Illumina. Methylation within the STING genomic regions was determined by selecting probes within 10 KB of the STING transcription start site. Copy Number Variant (CNV) profiles were derived from methylation signals using the cnSegmentation() function from SeSAMe. Comparison glioblastoma methylation data ($n = 170$ patients) was sourced from the TCGA database and processed using the same methods[89]. Publicly available meningioma methylation data was also incorporated with the institutional cohort, also processed using the same methods ($n = 904$ patients)[2,7,11].

## Bulk RNA sequencing analysis

Expression of STING and other genes in meningiomas was quantified after harmonization of publicly available RNA-seq datasets for a total of 584 meningiomas with paired methylation profiles[2,11,13]. Following RNA purification, all samples were sequenced on the Illumina platform, using single- or paired-end sequencing, with read lengths of 50 bp and 150 bp. Raw FASTQ files were downloaded from all studies, and reads were processed using the standard ENCODE pipeline with initial adapted trimming using Trim Galore! (v.0.6.1) with the paired parameter. Subsequently, trimmed reads were mapped to genome (GRCh38) using STAR (v.2.7.5) and quantified using RSEM (v.1.3.1)[90,91]. Tximport was used to import RSEM counts into R (v.4.3.1), where data were initially processed into log counts using the edgeR pipeline[92,93].

Briefly, counts were first batch corrected for institution of origin and single/paired-end sequencing using ComBat-seq, and genes were filtered for low expression using standard edgeR parameters[94]. Then, counts were converted into counts per million (cpm) and then $\log_2$ transformed. Final preprocessing was applied to normalize effective library sizes (calcNormFactors() from edgeR) and transform data for linear modeling purposes (voom() from limma)[95]. Comparison of glioblastoma expression data with paired methylation data referenced in "Bulk DNA-methylation profiling and analysis" ($n = 170$ patients) was sourced from the TCGA database and processed using the same methods. An additional 4 samples of ex vivo, patient-derived meningioma cell cultures were also processed for RNA-seq, with each sample either being an untreated control or treated with 8803. Differential expression was performed on this data using limma-voom. Single-sample gene set enrichment analysis (ssGSEA) was performed using the GSVA package[96].

## Multiplex fluorescence staining

Multiplex profiling of paraffin-embedded meningioma slides (4 um thick) was performed using a sequential immunofluorescence (seqIF™) panel that included the following unconjugated antibodies: CD31 (endothelial cells; Abcam, clone EPR17259, cat#ab225883), GFAP (astrocytes in adjacent brain; Sigma, clone GA5, cat#MAB360), SSTR2 (meningioma tumor cells; Abcam, clone UMB1, cat#ab134152), CD163 (macrophage scavenger receptor; Abcam, clone EPR19518, cat#ab182422), STING (Cell signaling, clone D2P2F, cat#13647), and p-IRF3 (downstream activation of the STING pathway; Cell signaling, clone 4D4G, cat#4947). All antibodies were validated using conventional immunohistochemistry and/or immunofluorescence (IF) staining, in conjunction with the corresponding fluorophore and the spectral 4′,6-diamidino-2-pheynlindole (DAPI; ThermoFisher Scientific) counterstain. For optimal concentration and the best signal/noise ratio, all antibodies were tested at three different dilutions, starting with the manufacturer-recommended dilution (MRD), then MRD/2 and MRD/4. Secondary Alexa Fluor 555 (ThermoFisher Scientific) and Alexa Fluor 647 (ThermoFisher Scientific) were used at 1/200 and 1/400 dilutions, respectively. The optimizations and full runs of the multiplex panel were executed using the seqIF™ methodology integrated into the Lunaphore COMET™ platform (characterization 2 and 3 protocols, and seqIF™ protocols, respectively)[97]. The staining is performed using automated cycles of 2 antibodies at a time, followed by imaging and elution, in which no sample manipulation is required. All reagents were diluted in Multistaining Buffer (BU06, Lunaphore Technologies). The elution step lasted 2 minutes for each cycle and was performed with Elution Buffer (BU07-L, Lunaphore Technologies) at 37 °C. Quenching lasted for 30 sec and was performed with Quenching Buffer (BU08-L, Lunaphore Technologies). Staining incubation times lasted four minutes for all primary antibodies and two minutes for secondary antibodies. Imaging was performed in Imaging Buffer (BU09, Lunaphore Technologies) with an integrated epifluorescent microscope at 20x magnification. Image registration was performed immediately after concluding the staining and imaging procedures by COMET™ Control Software. Slides were automatically scanned using the Lunaphore COMET™ following the manual's instructions with a fluorescent high-power field scan after each staining cycle (20x magnification microscope). The microscope captures the fluorescent signals (DAPI, TRITC, and Cy5) separately at the corresponding fluorophore wavelength, with preset exposure times. Then, these captures are co-registered in one multilayer image (OME.TIFF) without disrupting the unique fluorescent spectral signature of the markers. Markers were analyzed and pseudo-colored using the HORIZON Viewer from Lunaphore.

## Ex vivo meningioma cell viability assay

Patient-derived meningiomas ($n = 9$ patients) were collected and processed into single-cell suspensions within an hour after resection. The selected samples encompassed a range of WHO grades, tumor locations, and DNA methylation subgroups. Tumor tissue was dissociated using collagenase (20 μl/ml; Roche) and DNase (1 μl/ml; Roche) in RPMI 1640 medium (Roswell Park Memorial Institute; Corning) supplemented with 2% fetal bovine serum (FBS). The cells were washed and then plated as $5 \times 10^5$ viable cells/well in triplicate. The cells were incubated for 24–72 h at 37 °C in Dulbecco's Modified Eagle Medium (DMEM) supplemented with GlutaMAX™ (Gibco) and 10% FBS before being treated with 8803 (10 μM). Tumor cell viability was measured using the MTT colorimetric assay kit (CellTiter 96® Aqueous One Solution Cell Proliferation Assay; Promega) at 24, 48, and 72 hours after treatment. Treated-group values were normalized to the control-group values set at 100% of the maximal signal. Differences between treated and untreated at the same time point were analyzed using 2-way ANOVA analysis.

## Immune cell removal of ex vivo meningioma cytotoxicity assays

Patient-derived WHO grade I meningiomas ($n = 3$ patients) were collected and processed into single-cell suspensions within an hour after resection. Tumor tissue was dissociated using collagenase (20 μl/ml; Roche) and DNase (1 μl/ml; Roche) in RPMI 1640 medium (Roswell Park Memorial Institute; Corning) supplemented with 2% fetal bovine serum (FBS). CD45+ and CD45- populations were isolated using CD45 selection columns (Miltenyi). The cell populations were washed, tagged with either green (CD45+ population) or red (CD45- population) cell tracker dyes (Thermofisher), and then plated separately or in combination (1:1 ratio) at $2.5 \times 10^5$ cells/well in triplicate. The cells were incubated for 2 hours at 37 °C in DMEM (Dulbecco's Modified Eagle Medium) supplemented with GlutaMAX™ (Gibco) and 10% FBS and then treated with 8803 (10 μM). After 48 hours, the cells were collected and stained for flow cytometry.

## Cell death pathway inhibition

The role of cell death pathways in mediating 8803 efficacy was evaluated via pharmacologic inhibition. Freshly resected meningiomas ($n = 3$ patients) underwent single-cell dissociation and were plated as described above. For each condition, tumor cells were plated in triplicate and then incubated for 24 hours to ensure baseline viability. Six to twelve hours before 8803 administration, an inhibitor from each of the following groups was added to the appropriate wells: apoptosis (pan-caspase Z-VAD-FMK inhibitor; 20 μM; Promega), necrosis (Necrostatin-1; 10 μM; Selleckchem), ferroptosis (Ferrostatin 1; 5 μM; MedChemExpress), pyroptosis (Disulfiram; 20 μM; Selleckchem), and autophagic cell death (3-Methyladenine; 5 mM; MedChemExpress). Cells were then treated with 8803, and viability was measured at 48–72 hours post-treatment via flow cytometry analysis. Differences between groups were analyzed using 2-way ANOVA analysis.

## Antioxidant assay

Meningioma cells were cultured in DMEM/F-12 media (Thermo Fisher, cat#10565-018) supplemented with 10% FBS (Cytiva, cat#SH3007103). A total of 5,000 cells per well were seeded in triplicate in 96-well plates. Glutathione-ethyl-ester (GSHee, Cayman, cat#14953) and N-acetylcysteine (NAC, Thermo Scientific, cat#160280250) were prepared as a 10 mM stock solution in PBS, filtered through a 0.22 μm PVDF membrane (Millipore Sigma, cat#SH3007103), and added to fresh media at the desired final concentrations. Cells were treated with 8803 at 10 ug in combination with each antioxidant. Cells were maintained in 100 μL of media and treated for 48 hours. Cell viability was assessed using the CellTiter-Glo® Luminescent Cell Viability Assay (Promega) according to the manufacturer's protocol. CellTiter-Glo® reagent (3 mL) was diluted in 9 mL of DPBS, and 100 μL of this solution was added to each well on top of culture media. Plates were shaken on an orbital shaker for 2 minutes and incubated at room temperature for 10 minutes.

Luminescence was measured using a Synergy 2 Microplate Reader (BioTek), and cell viability was calculated by normalizing to the luminescence values of untreated control wells.

## Meningioma cell lines

The IOMM-LEE (catalog number CRL-3370)[98] human meningioma cell line was obtained from the American Type Culture Collection and underwent pathogen and authentication testing by short-tandem repeat analysis before use in experiments. CH157-MN was obtained from Dr Craig Horbinski[99]. Both cell lines were propagated as monolayers in a complete medium consisting of Dulbecco's Modified Eagle Medium: Nutrient Mixture F-12 (DMEM/F-12, Gibco), 10% FBS (Cytiva; Marlborough, MA), and sodium pyruvate (Gibco). Half of the medium in each culture was replaced with fresh medium every 3–4 days. Culture passages were performed using Trypsin-EDTA (0.05%, Gibco, #25300-054). The murine MGS1 cell line was obtained from Dr. Michel Kalamarides (AP-HP Pitié-Salpêtrière University) and Dr. Jacky Yeung (Yale University) and was verified to be pathogen-free before implantation. This meningioma cell line was propagated in DMEM supplemented with 10% fetal bovine serum, 1% penicillin/streptomycin, and 1% L-glutamine, as previously reported[14,51]. All cells were incubated at 37 °C in a humidified atmosphere containing 95% $O_2$ and 5% $CO_2$.

## Western blot analysis of STING expression

CH157, IOMM-LEE, MGS1, GBM12, and 0827 cell pellets stored at -80 °C were thawed on ice, resuspended in cell lysis buffer (RIPA with 1X Halt protease inhibitor) and sonicated with a QSonica Q800R3 sonicator at 50% amplitude, 20 seconds on, 20 seconds off for a total of 40 seconds. Cell lysates were cleared of insoluble material by centrifugation and protein was quantified using a Bradford assay. The sample (25 ug/lane) was loaded onto a 4-12% Bis-Tris polyacrylamide gel (Cat# NP0323), run, and then transferred onto a 0.2 μm PVDF membrane. Membranes were incubated overnight with anti-STING (Cell Signaling, Cat# 50494, 1:1000), anti-GSDMD (Cell Signaling, Cat# 97558, 1:1000), or anti-GAPDH (Cell Signaling, Cat# 5174, 1:1000) primary antibodies, followed by incubation with an anti-rabbit HRP-conjugated secondary antibody (Cell Signaling, Cat# 7074, 1:10,000). Chemiluminescent imaging of blots was performed using the ChemiDoc MP imaging system (BioRad; Hercules, CA).

## Flow cytometry analysis

Primary meningioma tissue isolated from the operating room was enzymatically dissociated, and CD45+ cells were isolated using CD45 magnetic beads (Miltenyi). The CD45+ fraction was labeled with Cell-Tracker Green (Fisher), and the CD45− population was labeled with CellTracker Red (Fisher). Cells were then plated either alone or in combination and treated with 8803 for 48 hours, with or without cell death inhibitors. On the day of the assay, cells were lifted, washed with 1X PBS, and stained (1:1000) with eBioscience™ Fixable Viability Dye eFluor™ 780 for 30 minutes on ice. After subsequent washes, cells were resuspended in FACS buffer (PBS + 2% FBS). Samples were acquired using a FACSymphony A5-Laser Analyzer (Fig. S13), and data were analyzed using FlowJo software.

## Meningioma in vivo models

J:NU (strain 007850) and FVB/NJ (strain 001800) 6-8 weeks old mice were purchased from Jackson Laboratories (Bar Harbor, ME) for in vivo therapeutic evaluations of 8803.Male and female mice were used. All mice were housed under standard barrier conditions, which included filtered air and sterilized food, water, bedding, and cages and racks. Housing conditions included a 12-hour dark/12 light cycle, 18-23 °C ambient temperature and 40-60% humidity. The Northwestern University Institutional Animal Care and Use Committee (IACUC) approved all animal protocols, and all animal care followed institutional guidelines. For flank models, mice were bilaterally implanted

with 3 x 10⁶ IOMM-LEE cells. 8803 (5 ug) was delivered subcutaneously into the tumor using a Hamilton syringe, with treatment initiated as designated in the schemas. Tumors were measured with calipers in two directions. Flank tumor volumes were calculated using (LxW^2)/2. A total size of 3.0 cm in diameter of all tumors combined in mice was not exceeded per IACUC guidelines. Animals bearing tumors that interfered with normal functions, such as eating, ambulating, and eliminating waste, were considered humane endpoints. Rodents with functional impairments were immediately euthanized. Ulcerated tumors were considered an automatic endpoint. For orthotopic models, CH157 or IOMM-Lee cells ($1.0 \times 10^5$ cells) were implanted under the skull in the right frontal subdural region, approximately 2 mm posterior to the bregma and 1 mm to the right of midline, and to a depth of 3 mm from the bone surface. 8803 treatment (5 ug) was delivered either through a previously implanted guide cannula or using the same stereotactic coordinates for the tumor implantation. Treatment initiation and schedule are designated in the schemas. For the immune-competent murine meningioma model, the tumorigenic dose for MGS cells was $2.5 \times 10^5$ in a total volume of 2.5 μl. Typically, five days after implantation, mice were randomized to either the 8803-treatment group or vehicle control. Animals were euthanized by $CO_2$ asphyxiation and general anesthesia when they became moribund (>20% weight loss, neurologic symptoms, or evidence of pain/distress), and survival time was recorded. Survival analyses were performed using GraphPad Prism 9 software.

## Sc-RNA sequencing of 8803 treated meningiomas

Six mice were utilized for sc-RNA experiments for each condition (treated; control). MGS1-bearing FVB/N mice were treated with 8803 on day 21 and euthanized on day 28. Tumors were grossly dissected and digested using collagenase (20 μl/ml; Roche) and DNase (1 μl/ml; Roche) in RPMI 1640 medium (Roswell Park Memorial Institute; Corning) supplemented with 2% fetal bovine serum (FBS). The CD45+ population was isolated using Percoll. Three mice were pooled in preparation for each sequencing run and spiked 1:1 with CD45- before sequencing to obtain sufficient immune cells for profiling. Sequencing was performed by the Northwestern University NUSeq sequencing facility core, with a targeted sequencing depth of 20,000 reads per sample acquired for RNA, and a total of 20,000 cells were targeted for each sample. CellRanger was used to align subsequent reads to mm10. Similar to the human meningioma data, the Seurat R Package using the scRNA-seq Seurat10x genomic workflow was used for all analyses[84]. SoupX was used to remove ambient RNA, and scDblFinder was used to remove doublets. Following this, cells were filtered using a percent mitochondrial DNA threshold of 10% and a UMI range of 100 to 4000. Cells were then subject to Log Normalize, Scale Data, and PCA functions. Murine tumor cells were identified through reference mapping to human meningioma cells. Briefly, human orthology of murine genes was called from Orthology.eg.db, following which Seurat function of FindTransferAnchors and TransferData was used to map murine query data onto human meningioma samples. Immune cells were subsequently subclustered, and the FindClusters and FindMarkers functions were utilized for clustering and marker identification. Non-linear dimensional reduction techniques were applied to visual data in UMAP plot format. The Harmony algorithm was used to regress batch effects[83]. The MiloR package was utilized for differential abundance testing using standard parameters and protocol. DEGs were calculated using pseudobulk analysis using the DESeq2 package between 8803 vs control.

## Murine macrophage culture and treatment

Bone marrow-derived macrophages were isolated from male and female mice from three strains of mice: FVB, MyD88, and Tmem173gt. Mice were euthanized, after which both hind limbs were collected and cleaned of muscle and connective tissue to expose the femur and tibia.

One end of each bone was cut, and the bones were placed cut-side down in a 0.6 mL microcentrifuge tube with a hole punctured at the bottom. This tube was placed within a 1.5 mL collection tube and centrifuged at $5000 \times g$ for 4 minutes to extract bone marrow. Isolated marrow was resuspended in Ammonium-Chloride-Potassium (ACK) lysis buffer and incubated for 3 minutes to lyse red blood cells. Lysis was quenched with media, and cells were counted and pelleted by centrifugation. The cell pellet was resuspended in complete media containing 40 ng/mL Macrophage Colony-Stimulating Factor (M-CSF). Two million macrophages were seeded per well in a 6-well plate and maintained in culture for one week, with fresh media added every three days. Simultaneously, MGS1 cells were seeded at $2 \times 10^6$ cells per 10 cm dish and cultured in parallel for preparation of the macrophage treatments. After culture, supernatant was collected and centrifuged at $2600 \times g$ for 5 minutes to remove cell debris. For lysate preparation, MGS1 cells were harvested by trypsinization and resuspended at $2 \times 10^6$ cells/mL in media. Cells were transferred to 0.5 mL sonication tubes and lysed using a Q800R3 Sonicator (QSonica, 3 cycles at 50% amplitude, 20 seconds ON, 20 seconds OFF, at 4 °C). The resulting lysate was centrifuged at $2600 \times g$ for 5 minutes to clear debris. Macrophages were treated with either MGS1-conditioned media or cell lysate for 48 hours. Following treatment, macrophage supernatants were collected and centrifuged to remove residual debris. Cleared supernatants were stored at –80 °C until analysis using a Luminex Multiplex Assay.

## Luminex multiplex assay

A multiplex Luminex assay (R&D Systems, cat#LXSAMSM) was performed according to the manufacturer's instructions. Macrophage culture supernatants were diluted 1:2 in calibrator diluent (75 μL sample + 75 μL diluent) before analysis. Briefly, 50 μL of each diluted sample was incubated with a premixed magnetic microparticle cocktail in a 96-well plate for 2 hours at room temperature on a shaker (800 rpm), protected from light. Following three washes, biotinylated detection antibodies and streptavidin-PE were sequentially added, followed by incubation and washing. Plates were read on a Luminex® 100/200™ analyzer, and the median fluorescence intensity (MFI) was recorded for each analyte. MFI readings were converted to analyte concentrations using a 5-parameter logistic (5-PL) standard curve. Final values were adjusted based on the dilution.

## Electron microscopy

To obtain Transmission Electron Microscopy (TEM) data, cultured meningioma cells were fixed in glutaraldehyde (2.5%) and paraformaldehyde (2%) in a cacodylate buffer (0.1 M) overnight at 4 °C. After fixation in 1% osmium tetroxide and 3% uranyl acetate, meningioma cells were dehydrated using ethanol, then embedded in Epon resin and polymerized for 48 hours at 60 °C. Ultrathin sections were cut using an Ultracut UC7 Ultramicrotome (Leica Microsystems, Germany) and contrasted with 3% uranyl acetate and Reynolds's lead citrate. Imaging of the samples was performed on a FEI Tecnai Spirit G2 transmission electron microscope (FEI Company, Hillsboro, OR) operated at 80 kV. An Eagle 4k HR 200 kV CCD camera was used for image capture. For Scanning Electron Microscopy (SEM), the medium was removed after cell incubation, and fixation was performed overnight at 4 °C in glutaraldehyde solution (2%). After rinsing of the cells three times in 0.1 M sodium cacodylate buffer, incubation was performed in osmium tetroxide (1%) for 2 h. Specimens were subsequently dehydrated in a series of ethanol washes (50% to 100%), critical-point-dried with carbon dioxide (Samdri-790, Tousimis, Rockville, USA), mounted on a carbon tape with an aluminum stud, and coated with 20 nm gold in a EM ACE600 Sputter coater (Leica, Germany). Finally, images were captured under a scanning electron microscope (JCM-6000PLUS, JEOL, Japan).

## Statistical analysis

Human patient sample size was based on feasibility of obtaining postoperative surgical resections specimens within a two-year time window. No data was excluded, and replications were confirmatory. Statistical significance was evaluated in pairwise comparisons with control values using a One-way and Two-way ANOVA test after confirmation of normal distributions with GraphPad Prism 8.0 and R (v 4.3.1). All samples were quantified as means + SEM, and $P < 0.05$ was considered significant. Interaction effects between conditions in the Luminex assay were modeled using linear regression in R. For survival endpoints in the preclinical studies, PASS 2021 indicates that a sample size of 10 mice/group achieves 80% power to detect a log hazard ratio of −0.69 (i.e., hazard ratio 0.5) at a 0.025 significance level using the Cox proportional hazard model. Mice were randomized into experimental treatment arms. The endpoint of death did not require blinding.

## Reporting summary

Further information on research design is available in the Nature Portfolio Reporting Summary linked to this article.

## Data availability

All data used to support the findings of this study are available within this article or via public repositories. Single-nucleus multiome data (RNA + ATAC) generated in this study have been deposited in the NCBI Gene Expression Omnibus (https://www.ncbi.nlm.nih.gov/geo/) under accession number GEO GSE297758. Single-cell RNA-seq of treated and control mice is deposited under GEO GSE297764 (super series) (https://www.omicsdi.org/dataset/geo/GSE297764). Bulk RNA-seq of ex vivo-treated samples is deposited under GEO GSE297637. Previously reported human scRNA-seq data is available under accession numbers GSE183655 and PRJNA826269[2,3]. Bulk DNA methylation was analyzed from datasets GSE189521, GSE183647, and GSE221029[2,7,11]. Meningioma and glioblastoma bulk RNA-seq data were obtained from GSE189672, GSE183653, and GSE212666. Further source data are provided in this paper under supporting data values. Previously reported multiome data (RNA + ATAC) for high-grade glioma are available at GSE230389[100].

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

## Acknowledgements

Lunaphore COMET™ multiplex immunofluorescence was enabled by a gracious gift from the Stephen M. Coffman Trust to the Northwestern Medicine Malnati Brain Tumor Institute of the Lurie Cancer Center. Multiplex assays using technology from Luminex were generously performed by the Comprehensive Metabolic Core at Northwestern University. Electron microscopy work was performed at the Northwestern University Center for Advanced Microscopy generously supported by NCI CCSG P30 CA060553 awarded to the Robert H Lurie Comprehensive Cancer Center. This study was supported by NIH grants CA120813, NS120547, P30CA060553, P50CA221747, the Remission Alliance, the Charlie Teo Foundation, Robert Mosky, and gracious funding from the Malnati Brain Tumor Institute to ABH; the American Brain Tumor Association (fully supported by Tap Cancer Out), US Department of Defense CA230856, and the Neurosurgeon Scientist Training Program of the Society of Neurological Surgeons to MWY; NIH/NCI Training Grant T32CA009560 to ST.

## Author contributions

Conceptualization: M.W.Y., S.T., H.N., and A.B.H. Data Curation: M.W.Y., S.T., H.N., H.C., L.K.B., C.S., D.T., R.C., Q.P., C.M.W., V.C., Alicia S., D.O., L.F., A.S., J.F., M.H., S.J.T., C.M.H., J.T.A., P.J., D.J.B., R.V.L., M.S.L., A.M.S., J.P.C., M.C.T., and S.T.M. Resources: A.C., D.O., L.F., A.S., J.F., M.H., S.J.T., J.Y., M.P., M.K., P.Z., A.H.S., D.M.A., C.M.H., R.S., M.S.L., A.M.S., J.P.C., M.C.T., S.T.M., J.M., G.P.D., M.A.C., and D.R.R. Formal Analysis: M.W.Y., S.T., H.N., H.C., M.G., and T.K.S. Supervision: A.B.H. Funding Acquisition: M.W.Y., M.S.L., S.T.M. and A.B.H. Investigation: M.W.Y., S.T., H.N., H.C., M.G., T.K.S., L.H., W.W., N.S., R.C., Alicia S., and K.M. Visualization: M.W.Y., S.T., H.N., H.C., and Q.P. Writing – Original Draft: M.W.Y., S.T., H.N., and A.B.H. Writing – Review and Editing: All authors.

## Competing interests

MAC declares the following competing interests: grants and personal fees from ImmunoGenesis that are relevant to the presented study, and unrelated work from Alligator Bioscience, ImmunOS, Oncoresponse, Kineta, Xencor, Astrazeneca. MAC holds the patent Cyclic dinucleotides as agonists of stimulator of interferon gene dependent signaling licensed to ImmunoGenesis, Inc. The remaining authors declare no potential competing interests related to the content of this manuscript.

## Additional information

Mark W. Youngblood [1,2,16], Shashwat Tripathi[1,2,16], Hinda Najem [1,2,16], Harrshavasan Congivaram[1,2], Mateo Gomez[1,2], Thomas K. Sears [1,2], Lisa Hurley[1,2], Leah K. Billingham[1,2], Caylee Silvers[1,2], Wenxia Wang[1,2], Deanna Tiek [1,2], Nishanth Sadagopan[1,2], Rahul Chaliparambil [1,2], Qianyi Pu[1,2], Ching Man Wai[3], Abrar Choudhury [4], Alicia Steffens[1,2], Kathleen McCortney [1,2], Gustavo Ignacio Vazquez Cervantes[1,2], Daniel Oyon[1], Luis Fernandez[1], Ashley Selner[1], Jawad Fares[1], Matthew Hagan[1], S. Joy Trybula[1], Jacky Yeung [5], Matthieu Peyre [6], Michel Kalamarides[6], Peng Zhang [1,2], Alexander H. Stegh [7,8], David M. Ashley [9,10], Craig M. Horbinski [1,2,11], Jared T. Ahrendsen [2,11], Pouya Jamshidi [1,2,11], Daniel J. Brat[1,2,11], Rimas V. Lukas[1,2], Roger Stupp[1,2], Maciej S. Lesniak [1,2], Adam M. Sonabend [1,2], Gavin P. Dunn[12], James P. Chandler[1,2], Matthew C. Tate[1,2], Stephen T. Magill [1,2], Jason Miska [1,2], Michael A. Curran [13], David R. Raleigh [4,14,15] & Amy B. Heimberger [1,2] ✉

[1]Department of Neurological Surgery, Feinberg School of Medicine, Northwestern University, Chicago, IL 60611, USA. [2]Malnati Brain Tumor Institute of the Robert H. Lurie Comprehensive Cancer Center, Feinberg School of Medicine, Northwestern University, Chicago, IL 60611, USA. [3]Northwestern University Sequencing Core, Chicago, IL, USA. [4]Department of Radiation Oncology, University of California San Francisco, San Francisco, CA, USA. [5]Department of

Neurosurgery, Yale School of Medicine, New Haven, CT, USA. [6]Department of Neurosurgery, Hospital Pitie-Salpetriere, AP-HP & Sorbonne Université, F-75103 Paris, France. [7]Taylor Family Department of Neurological Surgery, Washington University School of Medicine, St. Louis, MO, USA. [8]The Brain Tumor Center, Washington University School of Medicine/Alvin J. Siteman Comprehensive Cancer Center, St. Louis, MI, USA. [9]Department of Neurological Surgery, Duke University Medical School, Durham, NC 27710, USA. [10]Preston Robert Tisch Brain Tumor Center, Duke University Medical School, Durham, NC 27710, USA. [11]Department of Pathology, Feinberg School of Medicine, Northwestern University, Chicago, IL, USA. [12]Department of Neurosurgery, Massachusetts General Hospital, Boston, MA, USA. [13]Department of Immunology, University of Texas MD Anderson Cancer Center, Houston, TX 77030, USA. [14]Department of Neurological Surgery, University of California, San Francisco, CA, USA. [15]Department of Pathology, University of California, San Francisco, CA, USA. [16]These authors contributed equally: Mark W. Youngblood, Shashwat Tripathi, Hinda Najem. ✉e-mail: amy.heimberger@northwestern.edu

