## [Transparent Peer Review file · Nature Communications]

STING activation induces cytotoxic and immune responses in meningiomas via mitochondria-mediated programmed necrosis

Corresponding Author: Professor Amy Heimberger

Version 0:

Reviewer comments:

Reviewer #2

(Remarks to the Author)

The authors have addressed the majority of my previous comments experimentally and some through acknowledgement of the limitations and caveats in the discussion section. I appreciate the efforts the authors have put into this revision and support its publication.

Reviewer #4

(Remarks to the Author)

The authors have sufficiently addressed the comments raised by both Reviewer #1 and Reviewer #3. The methodological concerns have been handled appropriately through clear quality control, batch management, and additional analyses. The figure legends point readers to the relevant methods and supporting analyses. While the sample imbalance between conditions could be a potential confounding factor, the authors' use of MiloR, which is widely applied to mitigate cell-number imbalance, appears appropriate and yielded robust and reliable results. In addition, the number of mice per condition in Figure 5 seems adequate to support the conclusions, particularly given the consistency with the human data.

It would nevertheless be interesting to identify which macrophage subsets differ between the control and 8803. Moreover, a recent single-cell study of matched primary and recurrent meningiomas demonstrated enrichments of one of the collagen family members, COL6A3, in recurrent tumors. It would be interesting to know whether COL6A3 enrichment is also observed in the current cohort. Lastly, to improve the reproducibility of cell-type annotation, I recommend providing a comprehensive marker table used for cell-type assignments. While not mandatory, the current color scale gradient, where high gene expression is shown in blue and low expression in red, can be confusing as it differs from conventional use. This likely contributed to Reviewer #3's concern regarding Figure 3F.

Reviewer Comments

We thank the reviewers for their thoughtful comments and suggestions. To address these, the following new experimental data has been generated to support the findings of this manuscript:

- Longitudinal scanning electron microscopy of meningiomas treated with 8803 demonstrating membrane pore formation that compromises cellular integrity (new Fig. 3J; Fig. S7), thereby confirming direct cytotoxic effects on the meningioma cell that do not involve the immune system.
- Transmission electron microscopy showing that 8803 induces mitochondrial cristae collapse and compromise of the double membrane (new Fig. 3L; Fig. S7), providing additional data regarding the mechanism for direct cell death.
- In vivo treatment of meningioma at a later time (day 17) that compares control versus 8803 (new Fig. 4F).

Reviewer #1:

1. General statement

The manuscript presents STING induction as a promising therapeutic strategy for meningiomas, supported by a plethora of both ex vivo and in vivo functional and mechanistic data. These functional findings are compelling and well-supported. However, the accompanying multi-omic characterization appears underdeveloped, both in terms of analytical depth and sample size. Moreover, it often feels disconnected from the central narrative. To enhance clarity and impact, I recommend bringing the functional data to the forefront and moderating conclusions drawn from the limited single-cell datasets.

Revision: The multi-omic characterization depth has been increased throughout the manuscript. The sample size was incorrectly described and has been corrected. The narrative has been revised throughout the manuscript to increase the central narrative, and additional functional data has been added.

2. Figure Legends and Data Clarity

Across the manuscript, the figure legends lack sufficient methodological detail and often focus on interpretation rather than explanation. This makes it difficult to assess the analysis pipelines and understand how results were derived. I recommend revising figure legends to clearly state: data sources, sample sizes, statistical tests.

Revision: To improve clarity and reproducibility, these details for each analysis have been provided in Figure Legends 1A, 1C, 1D, 1E, 1F, 1G, 1H, 2A, 3G, 5D, 5E, 5G, and 5H). Furthermore, the specific R functions utilized in the creation of the data are displayed in each panel. This information is also cross-referenced in the corresponding methods sections.

3. Statistical Considerations

Key single-cell analyses, including comparisons of cell-type frequencies and gene expression (e.g., Figures 1C, 1G, 2F-H), should be aggregated and statistically evaluated at the level of biological replicates (patients or mice) rather than individual cells. Many of the results are presented without appropriate statistical testing. Of particular concern are the single-cell studies in Figure 5, which are based on only two replicates ($n=2$) and are therefore significantly

underpowered. Conclusions from these analyses should be toned down to reflect their exploratory nature.

Response and Revisions: Statistical comparisons performed in panel 1C were created using MiloR differential abundance testing, which is performed at the level of biological replicates. The points displayed represent “neighborhoods,” not individual cells. In the legend we clarify: *“Differential abundance testing was conducted on batch corrected sample level clusters using MiloR standard workflow including: buildgraph, makeNhoods, countcells, calcNhoodDistance, and testNhoods. Visualization was performed using plotDAbeeswarm. Neighborhoods are colored if p-value < 0.05 otherwise shown in grey. Immune cell populations with fewer than 5 significant neighborhoods were removed from the strip plot. Data was also analyzed using t-tests (with multiple testing correction) and results are further plotted in Fig. S2.”* In Figure S2, we display this data as box plots where each point represents a single patient having performed multiple corrections.

In panel 1G, we agree that this should have been done at sample-size, although we initially conducted an exploratory analysis hence using Wilcoxon Rank Test through FindMarkers. We have provided a new sample-level pseudobulk volcano plot using DESeq2. The main conclusions remain the same. Additionally, the following clarification was made in the legend: *“Volcano plot showing select chemokines enriched in the dura (left side) and meningioma tissues (right side) based on scRNA-seq of immune clusters shown in panel A. The annotated chemokines correlate with the elevated myeloid (blue text) and lymphoid (red text) populations in the meningioma and dura, respectively. Differentially expressed genes were calculated through pseudobulk analysis using DESeq2 analysis on immune cells grouped to compare between dura and meningioma.”*

Since the manuscript did not contain a submitted Fig. 2F-H we are presuming that the reviewer is referring to Fig. 5F-H. The n represented the number of sequencing runs rather than the number of mice (n=12). Figure Legend 5D has been corrected: *“UMAP plot of scRNA-seq data of 42,150 immune cells (CD45+). N = 6 mice per condition (untreated and 8803). Three mice were pooled for each sequencing run.”* Since the data represents n = 2 sequencing run and for enhance data clarity, we have provided DEGs using both Wilcoxon Rank Test through FindMarkers and DESeq2 sample level. In Fig. 5G we clarified: *“Differentially expressed genes were calculated using Seurat FindMarkers using Wilcoxon Rank Test between 8803 vs control from immune cells. Data was also analyzed using DESeq2 and presented in Fig. S8D.”* In Fig. 5H, the following details are provided: *“Differentially expressed genes were calculated using Seurat FindMarkers using Wilcoxon Rank Test between 8803 and control from meningioma tumor cells. Data was also analyzed using DESeq2 and presented in Fig. S8E.”*

4. Data Availability and Reproducibility

While the authors reference accession numbers for external datasets, they should also provide annotated and integrated data objects (e.g., Seurat objects) to facilitate reproducibility and allow the community to explore the integrated analyses directly.

Response: These files were submitted/uploaded on 10/3/2025.

Figure 1: This figure begins with a reanalysis of previously published single-cell transcriptomic data, but its connection to the rest of the manuscript remains unclear. The added value beyond the original studies is unclear, as STING appears to a promising targeting strategy, while its baseline activity seems not being correlated with any clinical outcomes. If this analysis does not inform subsequent findings, the authors should consider citing the sources instead of reproducing it.

Response and Revision: Although prior published studies have characterized the immune infiltration of meningiomas, the analysis conducted in Fig. 1 demonstrates for the first time that collagen-mediated immune suppression, including LAIR-1, predominates within the meningioma microenvironment which has not been previously described. These data lay the groundwork for Fig. 5 in which 8803 is shown to reverse this dominant immune suppressive mechanism. To help clarify this point, the header for the Results section has been modified: “*Single-cell and spatial analyses identify collagen-mediated immune suppression as a predominant feature of the meningioma microenvironment.*”

The integration of multiple published datasets lacks critical validation steps. Basic quality control metrics and integration diagnostics (e.g., batch effect correction, patient-level summaries) are not shown.

Revision: The new Fig. S1 includes batch effect correction, nFeature and nCount violin plots, and patient-level summaries.

Where possible, summary statistics at the patient level and appropriate statistical tests should be provided.

Response and Revision: Differential abundance at the patient level in form of box plots is provided in Fig. S2 corresponding to MiloR plot shown in Fig 1B and C. Otherwise, all analyses included have been performed on the patient/mouse level.

Figure 3: The color coding in this figure is not explained and should be clarified.

Revision: We have clarified this in the legend: “*Volcano plot showing select chemokines enriched in the dura (left side) and meningioma tissues (right side) based on scRNA-seq of immune clusters shown in panel A. The annotated chemokines correlate with the elevated myeloid (blue text) and lymphoid (red text) populations in the meningioma and dura, respectively. Differentially expressed genes were calculated through pseudobulk analysis using DESeq2 analysis on immune cells grouped to compare between dura and meningioma.*”

• Can the authors address the observed inter-patient heterogeneity for blocking apoptosis (Fig. 3D)? What distinguishes responders from non-responders to STING agonists in this context?

Response and revision: Response biomarkers for immune therapeutics are a significant challenge for the entire field of oncology, even for well-established therapeutics like anti-PD-1. Given the complexity of orchestrating multiple steps of immune activation and cytolytic activity, it is unlikely something simple will be predictive of therapeutic response to 8803. As an example of this type of challenge, one of the few FDA-approved biomarkers for immune checkpoint inhibition, tumor mutation burden is not predictive of therapeutic response across cancer lineages (PMID: 33736924). Factors that were analyzed to distinguish responders versus non-responders for 8803 included STING promoter methylation, global methylation (every CpG), and methylation subgroup as a linear regression against percent viability. None correlated with response. As such, response to 8803 will almost certainly be multi-factorial: “*Because meningiomas generally demonstrate chromosomal stability and low basal levels of STING activation, the STING pathway may be primed to respond to an agonist, thereby yielding tumor control. Most likely these responses are based on a variety of factors including epigenetic regulation of the STING promoter, STING isoforms and mutations, immune composition of the TME, and other regulatory factors that are now being elucidated. To investigate the therapeutic potential of this pathway in meningiomas, we used the agonist 8803, a 2',3'-thiophosphate CDN analog that we previously*

designed, which shows superior and robust activation of the STING pathway based on benchmarking to ADU-S100. While therapeutic resistance to STING agonists can be acquired through mutations and alterations of STING isoforms that render them insensitive to stimulation, 8803 can still generate immune activation in this scenario. Further biomarker response clarification studies will be conducted in parallel with the Phase I clinical trial.

Figure 4: The data show convincingly that early administration of STING agonists yields a strong response, using multiple controls. However, at later time point, comparable controls are not included, weakening this conclusion. A direct comparison between treated and vehicle groups at late time points would strengthen the argument considerably.

Revision: A direct comparison of 8803 and PBS treated controls at a later point (day 17) is provided in Fig. 4F. This day was selected since it represents the last feasible time point with enough time to measure both control and 8803-treated tumors before becoming necrotic with skin breakdown which requires the experiment to be terminated.

Figure 5: As noted before, the single-cell analysis from this figure is underpowered (n=2). Final interpretations should be made at the biological replicate level. Either more mice should be analyzed, or the conclusions must be framed as exploratory.

Response and revision: The n represented the number of sequencing runs rather than the number of mice (n=12). Figure Legend 5D has been corrected: *“UMAP plot of scRNA-seq data of 42,150 immune cells (CD45+). N = 6 mice per condition (untreated and 8803). Three mice were pooled for each sequencing run.”*

- Panel 5D: Please include split UMAP plots showing treated vs. control groups separately.

Revision: A split batch-corrected UMAP plot is provided in Fig. S8B.

- Panel 5F: Provide mean cell-type frequencies per murine replicate.

Response and Revision: This information is included in Fig. S8C; however, due to the sequencing run size of 2 per condition as the reviewer mentioned above, MiloR differential abundance testing was used to make conclusions regarding changes in the immune populations following 8803 administration.

- Panels 5G–5H: The figure legends do not specify the source or statistical treatment of these data. If these plots are derived from the same single-cell dataset, statistical analysis must be performed at the sample level, not the individual cell level. If other sources were used, please clarify explicitly.

Revision: The following clarifications in the Legends were provided: *“(G) Volcano plot of DEGs from immune cells showing that cytotoxic genes such as Prf1 and Ifng and matrix metalloproteinases are highly upregulated, whereas the immune suppressive molecules like TIM3 and LAIR are decreased following 8803 treatment. Differentially expressed genes were calculated using Seurat FindMarkers using Wilcoxon Rank Test between 8803 and control from immune cells. Data was also analyzed using DESeq2 and presented in Fig. S8D. (H) Volcano plot of DEGs from meningioma cells showing a decrease in several notable collagens and an increase in STING (Tmem173), GSDMD, and cell death caspases. Differentially expressed genes were calculated using Seurat FindMarkers using Wilcoxon Rank Test between 8803 vs control from meningioma tumor cells. Data was also analyzed using DESeq2 and presented in Fig. S8E.”*

- The transcriptional downregulation of collagens and upregulation of MMPs following STING agonist treatment is intriguing. What is the proposed mechanistic basis for these changes? Additional discussion or referencing of supporting literature would be helpful.

Response and Revision: This has not been described previously in the literature. In the Discussion, we provided the mechanistic basis for this finding based on the generated data. *“Notably, some cytokines like MMP were specifically induced by DAMPs, while others were more strongly activated by 8803 directly, thus demonstrating the involvement of both the TLR and STING pathways in triggering inflammation in the setting of direct tumor cytotoxicity via 8803. The immunocompetent MGS1 models only grows in the FVB/N background. As such, implantation of MGS1 into backgrounds that have genetically modified alterations in collagen or metalloproteinases will not support tumor growth. To overcome this limitation, we used macrophages from TLR and STING KO backgrounds that were treated with 8803 and/or meningioma lysate. The data indicates that collagen degrading MMP-3 is induced with the MGS1 lysate, which is abrogated in the MyD88 KO macrophages indicating that this effect requires the TLR pathway. 8803 alone does not trigger MMP-3 indicating meningioma cell death facilitates collagen degradation.”*

Overall, the manuscript is built on a strong foundation of functional data, and STING agonism appears to be a promising therapeutic approach in meningiomas. However, to improve scientific rigor and clarity, the authors should consider to emphasize functional results as the central narrative, revise the multi-omic analysis to reflect its exploratory nature and add statistical rigor, particularly for the partly underpowered single-cell data.

Reviewer #2:

In this manuscript, Youngblood et al. describe the expression and therapeutic targeting of the STING pathway in meningiomas. Using integrated transcriptional, epigenetic, and spatial approaches, the authors show broad expression of STING across multiple cell types in meningiomas, linking promoter hypomethylation and open chromatin accessibility to elevated STING expression. Importantly, they demonstrate direct tumor cell cytotoxicity mediated by programmed necrotic cell death pathways (pyroptosis, necroptosis, ferroptosis) upon exposure to the STING agonist 8803, independent of immune cell activity. This cytotoxicity involves ROS induction, caspase activation, and gasdermin D-mediated membrane disruption, leading to macrophage activation and extracellular matrix remodeling, especially collagen degradation, thus reshaping the immunosuppressive tumor microenvironment. These observations are complemented by robust in vivo data across several murine models, showing clear therapeutic efficacy of the agonist with significant tumor volume reduction, immune infiltration, and survival benefits. Overall, the study presents a comprehensive preclinical evaluation of STING agonism in meningiomas and proposes a unique therapeutic angle leveraging direct tumor cell cytotoxicity in addition to classical immune activation. Having said that, there are multiple key weaknesses and concerns regarding the novelty discussed below with suggested approaches to address them.

Novelty of the Therapeutic Approach: While the study demonstrates robust preclinical efficacy of STING agonism, the manuscript does not adequately address previous clinical setbacks with STING agonists despite compelling pre-clinical results. The authors need to articulate clearly why meningiomas represent a unique or particularly suitable context for STING activation relative to other solid tumors, particularly given multiple failed clinical trials with STING agonists. One plausible hypothesis, partially hinted at but insufficiently developed, is the relative chromosomal

stability of meningiomas and potentially lower basal STING pathway activation at baseline. The absence of pre-treatment chronic STING activation suggests that the pathway is primed to respond to an agonist through a pro-inflammatory signal that would yield tumor control. Demonstrating that meningiomas uniquely harbor STING expression without chronic activation would strengthen their claim to novelty.

Revision: In the Discussion, we clarify: *“The first clinical trial of a STING agonist for solid cancer used the ADU-S100 compound, which was injected directly into a solid tumor in patients with metastatic disease. Although well tolerated, with the injected lesions showing either stability or regression, there were only a few patients with partial responses. Other STING agonists such as BI 1703880, MK-2118, E7766, and IMSA101 have also been developed and evaluated in Phase I clinical trials, but the signals of response have remained modest. In contrast to these previously investigated indications, meningioma is usually a local, solitary disease. Because meningiomas generally demonstrate chromosomal stability and low basal levels of STING activation, the STING pathway may be primed to respond to an agonist, thereby yielding tumor control. Most likely these responses are based on a variety of factors including epigenetic regulation of the STING promoter, STING isoforms and mutations, immune composition of the TME, and other regulatory factors that are now being elucidated. To investigate the therapeutic potential of this pathway in meningiomas, we used the agonist 8803, a 2',3'-thiophosphate CDN analog that we previously designed, which shows superior and robust activation of the STING pathway based on benchmarking to ADU-S100. While therapeutic resistance to STING agonists can be acquired through mutations and alterations of STING isoforms that render them insensitive to stimulation, 8803 can still generate immune activation in this scenario. As such, the combination of 8803 in meningiomas represents a unique therapeutic opportunity that holds promise relative to previous clinical indications.”*

Pro-tumorigenic Roles of STING: The manuscript overlooks established evidence that STING activation also can promote metastasis, immune suppression, and tumor growth. Given that STING agonism is a double-edged sword, the authors must explicitly reconcile their findings with the literature describing STING as potentially pro-tumorigenic under chronic conditions. Clarification of acute versus chronic STING activation states in meningiomas would substantially enhance the manuscript's conceptual rigor. This might also bear on the clinical development path of STING agonists and how frequently it should be dosed in patients to maximize its efficacious effect while minimizing the side effects.

Revision: In the Discussion, we clarify: *“Though a previous study suggested that long-term STING induction may promote tumor growth, acute administration of 8803 in an immunocompetent setting is expected to be safe and efficacious. Indeed, STING agonists have not shown tumor-promoting effects across a variety of preclinical models, nor has this been reported in clinical trials. Pharmacological modulation using low pulsed dosing with STING agonists is critical for boosting T cell and antigen-presenting responses without inducing cell cycle-limiting and immunosuppressive feedback due to the effects of excess IFN signaling and high levels of neutrophil activation. Nonetheless, radiographic imaging of meningiomas in clinical studies using 8803 should more frequently assess for the possibility of tumor promoting effects although this would be unlikely given the intermittent nature of therapeutic delivery.”*

Immune Independence and Direct Cytotoxicity: A key strength of this study is the demonstration of STING agonist-induced direct cytotoxicity in tumor cells independently of immune cells, which is a significant departure from classical STING agonist applications. However, the authors need to provide clearer mechanistic evidence distinguishing the direct cytotoxic pathways from immune-mediated mechanisms.

Revision: New longitudinal scanning electron microscopy of only meningioma cells treated with 8803 demonstrate that the mechanism of direct cytolytic death is through the generation of membrane pores compromising cellular integrity (new Fig. 3J; Fig. S7). Transmission electron microscopy shows that 8803 directly induces mitochondrial cristae collapse and compromise of the double membrane in meningioma cells (new Fig. 3L; Fig. S7). In the Discussion, we have included the following clarification: “8803-induced STING activation triggers Gasdermin D, which damages both the membranes of the cell and mitochondria. This permeabilization leads to reduced mitochondrial membrane potential, increased ROS, and the release of mitochondrial DNA. The mitochondrial DNA leakage into the cytoplasm of the cells triggers additional STING activation through cGAS, thereby amplifying the response. Mitochondrial damage and the production of ROS are essential for mediating other types of cell death, including necroptosis, pyroptosis, and ferroptosis.”

Also why meningioma is an important question that should be addressed in this case, in light of the lack of clear clinical efficacy of STING agonists to date.

Revision: In the Discussion, we clarify: “The first clinical trial of a STING agonist for solid cancer used the ADU-S100 compound, which was injected directly into a solid tumor in patients with metastatic disease. Although well tolerated, with the injected lesions showing either stability or regression, there were only a few patients with partial responses. Other STING agonists such as BI 1703880, MK-2118, E7766, and IMSA101 have also been developed and evaluated in Phase I clinical trials, but the signals of response have remained modest. In contrast to these previously investigated indications, meningioma is usually a local, solitary disease. Because meningiomas generally demonstrate chromosomal stability and low basal levels of STING activation, the STING pathway may be primed to respond to an agonist, thereby yielding tumor control. Most likely these responses are based on a variety of factors including epigenetic regulation of the STING promoter, STING isoforms and mutations, immune composition of the TME, and other regulatory factors that are now being elucidated. To investigate the therapeutic potential of this pathway in meningiomas, we used the agonist 8803, a 2',3'-thiophosphate CDN analog that we previously designed, which shows superior and robust activation of the STING pathway based on benchmarking to ADU-S100. While therapeutic resistance to STING agonists can be acquired through mutations and alterations of STING isoforms that render them insensitive to stimulation, 8803 can still generate immune activation in this scenario. As such, the combination of 8803 and meningiomas represents a unique therapeutic opportunity that holds promise relative to previous clinical indications.”

Tumor Microenvironment Remodeling: The proposed mechanism whereby STING agonism remodels the collagen-rich, immunosuppressive microenvironment via DAMP release and macrophage activation is intriguing. However, the authors need more direct evidence linking the observed collagen reduction with therapeutic efficacy and macrophage activation. Specifically, genetic or pharmacological blockade of collagen degradation pathways (e.g., inhibition of matrix metalloproteinases) to determine the importance of this remodeling step would add considerable strength. Finally, and critically the authors need to rule in or rule out STING dependent type I interferon signaling in this process.

Response and Revision: In the Discussion: “The immunocompetent MGS1 models only grows in the FVB/N background. As such, implantation of MGS1 into backgrounds that have genetically modified alterations in collagen or metalloproteinases will not support tumor growth. To overcome this limitation, we used macrophages from TLR and STING KO backgrounds that were treated with 8803 and/or meningioma lysate. The data indicates that collagen degrading MMP-3 is

induced with the MGS1 lysate (Fig. 5), which is abrogated in the MyD88 KO macrophages (Fig. S10, Fig. S11) indicating that this effect requires the TLR pathway. 8803 alone does not trigger MMP-3 (Fig. S11) indicating that meningioma cell death facilitates collagen degradation.

Clinical Translation and Endpoint Considerations: The manuscript makes strong translational claims about the therapeutic potential of STING agonist 8803 for meningiomas, even suggesting radiographic tumor volume reduction as a clinical endpoint. It is unclear however if this means tumor eradication and whether this would translate into reduced recurrence in the clinical setting. Additional discussion of how these endpoints translate to clinically meaningful outcomes such as symptom relief, quality of life, or progression-free survival would provide valuable context for the clinical viability of this approach.

Response: These points are well taken, and we agree with the reviewer that they would need to be addressed in the context of a clinical trial. The pre-IND meeting with the FDA has been completed and the final IND enabling studies including CNS toxicity studies will be finished by the end of the year. The clinical trial endpoints that are being considered include longitudinal volumetric reduction or alterations of the growth curve on standard MRI imaging and neurological symptom alleviation. Exploratory endpoints that are being considered include decreased signal in ⁶⁸Ga-DOTATE PET post 8803 treatment to evaluate for decreased tumor signal and/or increased STING pathway activation engagement using [¹⁸F] FLT PET imaging (PMID: 34480004). We have touched upon this in a general way in the Discussion since these endpoints have not yet been finalized although many of the logistics have been worked out.

The manuscript could improve by clearer definitions and consistency in the descriptions of immune cell populations, particularly macrophage subsets. Figures and legends should explicitly clarify experimental conditions (e.g., dosages of agonist/inhibitors, exact animal numbers per group) to improve reproducibility.

Revisions and Response: To improve clarity and reproducibility, these details for each analysis have been provided in the figure legends (1A, 1C, 1D, 1E, 1F, 1G, 1H, 2A, 3D, 5D, 5E, 5G, and 5H). Furthermore, the specific R functions utilized in the creation of the data is displayed in each panel. This information is also cross-referenced in the corresponding methods sections

In conclusion, the manuscript is a strong preclinical contribution with exciting therapeutic implications. However, it requires critical refinement in addressing novelty concerns and mechanistic clarity to ensure the significance and uniqueness of STING agonism in meningiomas relative to existing literature and clinical experience.

Reviewer #3:

This study has systematically investigated the effect of STING expression and agonism in meningioma cells and their TME. Differences predicted by sc seq were validated by spectral Immunofluorescence staining. The investigators identify that STING is expressed in meningioma and these tumor cells were sensitive to sting agonist, with increased gene expression of pathways related to associated with autophagy, necroptosis, and proptosis. Thus treatment of mice bearing subcutaneous and orthotopic meningiomas with STING agonist 8803 showed anti-tumor efficacy. The study is well controlled, and the findings are significant, and support future translation of treating meningioma patients with STING agonist. The study while being translationally very significant stops short of mechanistically uncovering new pathways which advance our understanding of the role of STING pathway in meningioma and TME.

Figure 1: Move fig 1D to 1B. most highly differentially expressed genes/cluster that are used to assign cluster identity should be shown with Figure 1A, at the time of description of the clusters.

Response and Revision: We believe there was lack of clarity in our figure. Fig. 1A shows the highly differentially expressed genes that were used to assign cluster identity. This is also included in the Supplementary Materials as a full table with all DEGs. Fig. 1D shows select immune functional markers to interpret immunological functions in the context of meningioma and not used to assign lineage/cluster identifies. This has been clarified in the legend: “(A) Fifteen distinct immune populations (CD45 positive) were identified using single-cell RNA-sequencing data from meningioma (n = 13) and adjacent dura tissues (n = 9) based on the listed markers. Differentially expressed genes (DEGs) were found using standard parameters in Seurat *FindAllMarkers* (which use default non-parametric Wilcoxon rank sum test on all clusters), and examples are shown in each cluster box. 30,206 immune cells were included. (D) Gene expression dot plot of selected functional markers among the general immune populations shown in panel A. 30,206 total immune cells were used for analysis. Bubble size corresponds to the percent of cells expressing each marker; colors indicate average expression. Phagocytic TAMs have high expression of M2-like markers such as *CD163*, *CD204*, and *CD206*.”

Line 171: Markers for T.Tex and NK are not shown.

Response and Revision: The markers are shown in Fig. 1A. and this has been clarified in the text as such: “In contrast, terminally exhausted T cells (T.Tex; identified from markers highlighted in Fig. 1A) and NK cells were more frequently observed in meningioma-adjacent dura (Fig. 1B-C).”

Figure 1G: not clear if DEG genes are from bulk sequencing tumor Vs dura, or from select cell clusters in 1A. please clarify

Response and Revision: This has been clarified in the legend: (G) Volcano plot showing select chemokines enriched in the dura (left side) and meningioma tissues (right side) based on scRNA-seq of immune clusters shown in panel A. The annotated chemokines correlate with the elevated myeloid (blue text) and lymphoid (red text) populations in the meningioma and dura, respectively. Differentially expressed genes were calculated through pseudobulk analysis using DESeq2 analysis on immune cells grouped to compare between dura and meningioma.”

Legend of S4A says these pathways are up in sting high tumors, however the results does not say that. Is this analysis from sc seq shown in fig 1? How were sting hi Vs sting low tumors defined?

Response and Revision: The top related genes in this analysis are associated with the STING pathway. We did not designate STINGhi and low tumors but have provided further clarification in the legend.

Line 213: The authors state” Pseudo-bulk snATAC-seq data were used to analyze the cellular populations in meningioma (n = 4; 29,100 cells) and glioma (n = 2; 4,850 cells), which revealed consistent chromatin accessibility in the immune cell lineages between tumor types.” For pseudo bulk the single cells in each cluster are likely bulked together, so this was likely an expected result since clustering cannot be performed again.

Revision: In the Results, we clarify: *“Pseudo-bulk analysis snATAC-seq data in meningioma (n = 4; 29,100 cells) and glioma (n = 2; 4,850 cells) cells was performed to produce lineage specific ATAC tracks. Separate clustering of cancer-lineage specific immune populations revealed consistent chromatin accessibility in both tumor types.”*

Line 262 states: CASP1, 4, 7, 8, and 10 were 262 upregulated after treatment with 8803, but data in Fig 3F looks reduced after 8802 treatments.

Revision: Per the scale bar shown (bottom left), red values are low and blue values are high. The color designations for low and high values were harmonized throughout the entire manuscript with red representing down/decreased values as shown in Fig. 1D, 1G, 1H, 2D, 3F, 5G, and 5H.

Line 305: the authors state: Nude mice only have an intact innate immune system; these data indicate that the mechanism of action is not reliant on T cell-mediated immune responses. Since these mice don't express mature T cells, I think it is hard to make a statement about T cell involvement either way. This appears to be extrapolation of data. The authors can only state that the impact of T cells was not investigated.

Revision: This statement has been amended as follows: *“Because Nude mice do not express mature T cells, these data would suggest that the mechanism of action is not fully dependent on T cell-mediated immune responses and that 8803 may be exerting direct cytotoxicity in these in vivo models.”*

Reviewer Response Letter

It would nevertheless be interesting to identify which macrophage subsets differ between the control and 8803.

Revision: We have clarified that the tumor-associated macrophage (TAM) subtypes listed in the UMAP in the original Fig. 5D (new Fig. 7D) are macrophages. 8803 increased the infiltration of all three of the macrophage subtypes (ISG, Tissue Repair, and Phago) as shown in Fig. 5E and F (new Fig. 7E and F). The main text and figure legend have been adjusted to better reflect this enrichment of macrophage subtypes.

Moreover, a recent single-cell study of matched primary and recurrent meningiomas demonstrated enrichments of one of the collagen family members, COL6A3, in recurrent tumors. It would be interesting to know whether COL6A3 enrichment is also observed in the current cohort.

Response: As was noted in Supplementary Fig. S4, COL6A3 showed low expression within the included publicly available meningioma sequencing. These data were mostly confined to newly diagnosed meningioma, and as such, we are unable to analyze COL6A3 expression by recurrence status. We would direct the reader to PUBMED ID: 40593590

Lastly, to improve the reproducibility of cell-type annotation, I recommend providing a comprehensive marker table used for cell-type assignments.

Response and Revision: The cell type markers directly referenced in Fig. 1A for the human scRNA. Table S2 has been provided with the cell type annotation for murine scRNA.

While not mandatory, the current color scale gradient, where high gene expression is shown in blue and low expression in red, can be confusing as it differs from conventional use. This likely contributed to Reviewer #3's concern regarding Figure 3F.

Response: To keep consistency and avoid confusion with the rest of the panels throughout the entire manuscript where blue is high/upregulated and red is low/downregulated, we respectfully decline to change this specific panel.